# Learning with Optimized Random Features: Exponential Speedup by Quantum Machine Learning without Sparsity and Low-Rank Assumptions

**Hayata Yamasaki**
The University of Tokyo, Austrian Academy of Sciences
`hayata.yamasaki@gmail.com`

**Sathyawageeswar Subramanian**
University of Cambridge,
University of Warwick

**Sho Sonoda**
RIKEN AIP

**Masato Koashi**
The University of Tokyo

## Abstract

Kernel methods augmented with random features give scalable algorithms for learning from big data. But it has been computationally hard to sample random features according to a probability distribution that is optimized for the data, so as to minimize the required number of features for achieving the learning to a desired accuracy. Here, we develop a quantum algorithm for sampling from this optimized distribution over features, in runtime $O(D)$ that is linear in the dimension $D$ of the input data. Our algorithm achieves an exponential speedup in $D$ compared to any known classical algorithm for this sampling task. In contrast to existing quantum machine learning algorithms, our algorithm circumvents sparsity and low-rank assumptions and thus has wide applicability. We also show that the sampled features can be combined with regression by stochastic gradient descent to achieve the learning without canceling out our exponential speedup. Our algorithm based on sampling optimized random features leads to an accelerated framework for machine learning that takes advantage of quantum computers.

## 1 Introduction

Random features [1, 2] provide a powerful technique for scaling up kernel methods [3] applicable to various machine learning tasks, such as ridge regression [4], kernel learning [5], and principle component analysis [6]. Recently, Bach [7] has shown an optimized probability distribution of random features, and sampling features from this optimized distribution would drastically improve runtime of learning algorithms based on the random features. However, this sampling task has been computationally "hard in practice" [7] due to inversion of a high-dimensional operator. In contrast, the power of quantum computers to process data in quantum superposition attracts growing attention towards accelerating learning tasks, opening a new field: quantum machine learning (QML) [8–10]. In this work, we develop a framework of QML that accelerates a supervised learning task, by constructing an efficient quantum algorithm for sampling from this optimized distribution.

**Learning with random features:** Supervised learning deals with the problem of estimating an unknown function $y = f(x)$. We will consider $D$-dimensional input $x \in \mathbb{R}^D$ and real-valued output $y \in \mathbb{R}$. Given $N$ input-output pairs of examples, we want to learn $f$ to a desired accuracy $\epsilon > 0$. To model $f$, kernel methods use reproducing kernel Hilbert space (RKHS) associated with a symmetric, positive semidefinite function $k(x', x)$, *the kernel* [3]. Typical kernel methods that compute an $N \times N$ Gram matrix may not be scalable as $N$ gets large, but random features [1, 2], along with other techniques via low-rank matrix approximation [11–13], enable scalable kernel-based algorithms.

Algorithms using random features are based on the fact that we can represent any translation-invariant kernel $k$ as expectation of a feature map $\varphi(v, x) = \mathrm{e}^{-2\pi \mathrm{i} v \cdot x}$ over a probability measure $d\tau(v)$ corresponding to the kernel. Conventional algorithms using random features [1, 2] sample $M$ $D$-dimensional parameters $v_0, \ldots, v_{M-1} \in \mathbb{R}^D$ from the distribution $d\tau(v)$ to determine $M$ features $\varphi(v_m, \cdot)$ in time $O(MD)$. For a class of kernels such as Gaussian, this runtime may be reduced to $O(M \log D)$ [14, 15]. We learn the function $f$ using a linear combination of the $M$ features, i.e.,

$$f(x) \approx \sum_{m=0}^{M-1} \alpha_m \varphi(v_m, x) =: \hat{f}_{M, v_m, \alpha_m}(x). \tag{1}$$

To achieve the learning to accuracy $O(\epsilon)$, we need to sample a sufficiently large number $M$ of features. Once we fix $M$ features, we calculate coefficients $\alpha_m$ by linear (or ridge) regression to minimize an error between $f$ and $\hat{f}_{M, v_m, \alpha_m}$ using the $N$ given examples [2, 4, 16]. The sampling of features and the regression of coefficients can be performed simultaneously via doubly stochastic gradients [17].

**Problem:** These conventional algorithms using random features sampled from the data-independent distribution $d\tau(v)$ require a *large* number $M$ of features to learn the function $f$, which *slows down* the decision of all $M$ features and the regression over $M$ coefficients. To improve this, we aim to minimize $M$ required for the learning. Rather than sampling from $d\tau(v)$, we will sample features from a probability distribution that puts greater weight on *important features optimized for the data* via a probability density function $q(v)$ for $d\tau(v)$. To minimize $M$ achieving the accuracy $O(\epsilon)$, Bach [7] provides an optimized probability density function $q_\epsilon^*(v)$ for $d\tau(v)$ (see (3), Sec. 2.2). This optimized $q_\epsilon^*(v)$ achieves minimal $M$ up to a logarithmic gap among all algorithms using random features for accuracy $\epsilon$ [7]. It *significantly improves* $M$ compared to sampling from $d\tau(v)$ [4, 7, 18]; e.g., to achieve learning with the Gaussian kernel from data given according to a sub-Gaussian distribution, compared to sampling from the data-independent distribution $d\tau(v)$ in Refs. [1, 2], the required number $M$ of features sampled from the optimized distribution $q_\epsilon^*(v) d\tau(v)$ can be *exponentially small* in $\epsilon$ [7]. We call features sampled from $q_\epsilon^*(v) d\tau(v)$ *optimized random features*.

However, the sampling from $q_\epsilon^*(v) d\tau(v)$ has been "hard in practice" [7] for two reasons. First, the definition (3) of $q_\epsilon^*(v)$ includes an *infinite-dimensional operator* $(\Sigma + \epsilon \mathbb{1})^{-1}$ on the space of functions $f: \mathbb{R}^D \to \mathbb{R}$ with $D$-dimensional input data, which is intractable to calculate by computer without approximation. Second, even if we approximate $\Sigma + \epsilon \mathbb{1}$ by an operator on a finite-dimensional space, the *inverse operator* approximating $(\Sigma + \epsilon \mathbb{1})^{-1}$ is still hard to calculate; in particular, for achieving a desired accuracy in the approximation, the required dimension of this finite-dimensional space can be exponentially large in $D$, i.e., $O(\exp(D))$ [18, 19], and *no known algorithm* can calculate the inverse of the $O(\exp(D))$-dimensional operator in general *within sub-exponential time* in $D$.

Note that Refs. [20, 21] propose probability density functions similar to $q_\epsilon^*(v)$, from which the samples can be obtained in polynomial time [20–22]; however, in contrast to sampling from $q_\epsilon^*(v) d\tau(v)$, sampling from the distributions in Refs. [20, 21] does not necessarily minimize the required number $M$ of features for the learning. In particular, $q_\epsilon^*(v) d\tau(v)$ in Ref. [7] and the distribution in Ref. [20] are different in that the former is defined using an integral operator as shown in (3), but the latter is defined using the Gram matrix; even if we discretize the integral operator, we do not obtain the Gram matrix. The distribution in Ref. [21] does not use the integral operator either. Bach [7] proves optimality of $q_\epsilon^*(v) d\tau(v)$ in minimizing $M$ required for approximating function $f$, but this proof of the optimality is not applicable to the distributions in Refs. [20, 21]. Similarly, whereas sampling from an importance-weighted distribution may also be used in column sampling for scaling-up kernel methods via low-rank matrix approximation, algorithms in the setting of the column sampling [23–25] are not applicable to our setting of random features, as discussed in Ref. [7]. Quasi-Monte Carlo techniques [26, 27] also improve $M$, but it is unknown whether they can achieve minimal $M$.

**Our contributions:** As discussed above, the bottleneck in using random features sampled from the optimized distribution $q_\epsilon^*(v) d\tau(v)$ is each sampling step that works with inversion of $O(\exp(D))$-dimensional matrices for $D$-dimensional input data. To overcome this bottleneck and the difficulties in sampling from $q_\epsilon^*(v) d\tau(v)$, we discover that we can use a *quantum algorithm*, rather than conventional classical algorithms that run on existing computers. Our contributions are as follows.

- (Theorem 1) We construct a quantum algorithm for **sampling a feature from the *data-optimized* distribution** $q_\epsilon^*(v) d\tau(v)$ **in as fast as linear runtime** $O(D)$ **in the data dimension** $D$. The best existing classical algorithm for sampling each single feature from

*data-optimized* $q_\epsilon^*(v)d\tau(v)$ requires exponential runtime $O(\exp(D))$ [7, 18, 19]. In contrast, our quantum algorithm can sample the feature from $q_\epsilon^*(v)d\tau(v)$ in runtime $O(D)$, which is as fast as the conventional algorithms using random features [1, 2]. We emphasize that the conventional algorithms perform an easier task, i.e., sampling from a *data-independent* distribution $d\tau(v)$. Advantageously over the conventional algorithms sampling from $d\tau(v)$, we can use our algorithm sampling from $q_\epsilon^*(v)d\tau(v)$ to achieve learning with a significantly small number $M$ of features, which is proven to be *minimal* up to a logarithmic gap [7]. Remarkably, we achieve this without assuming sparsity or low rank of relevant operators.

- (Theorem 2) We show that we can combine $M$ features sampled by our algorithm with regression by stochastic gradient descent **to achieve supervised learning in time $O(MD)$, i.e., *without canceling out our exponential speedup*. This $M$ is minimal up to a logarithmic gap [7] since we use optimized random features. Thus, by improving the computational bottleneck faced by classical algorithms for sampling optimized random features, we provide a promising framework of quantum machine learning that leverages our $O(D)$ sampling algorithm to achieve the optimal $M$ among all algorithms using random features.

**Comparison with previous works on quantum machine learning (QML):** The novelty of our contributions is that we construct a QML algorithm that is *exponentially faster* than any existing classical algorithm sampling from $q_\epsilon^*(v)d\tau(v)$ [7, 18, 19], yet is still *free from sparsity and low-rank assumptions* on operators. Despite major efforts to apply QML to kernel methods [28], super-polynomial speedups like Shor's algorithm for prime factoring [29] are rare in QML. In fact, it has been challenging to find applications of quantum algorithms with super-polynomial speedups for practical problems [30]. Typical QML algorithms such as Refs. [31–34] may achieve exponential speedups over classical algorithms only if matrices involved in the algorithms are sparse; in particular, $n \times n$ matrices can have only $\mathrm{polylog}(n)$ nonzero elements in each row and column. Another class of QML algorithms such as Refs. [35–38] do not require sparsity but may attain large speedups only if the $n \times n$ matrices have low rank $\mathrm{polylog}(n)$. This class of quantum algorithms are polynomially faster than recent "quantum-inspired" classical algorithms such as Refs. [39–41], which also assume low rank. Quantum singular value transformation (QSVT) [42] has recently emerged as a fundamental subroutine to implement these quantum algorithms in a unified way. However, power and applicability of these QML algorithms are restricted by the extreme assumptions on sparsity and low rank [43].

Our key technical contribution is to develop an approach for circumventing the sparsity and low-rank assumptions in our QML algorithm, broadening the applicability of QML. We achieve this by combining the QSVT with another fundamental subroutine, quantum Fourier transform (QFT) [44, 45]. QFT and QSVT are commonly used in quantum computation [46, 47]; however, it is nontrivial to use these subroutines for developing a QML algorithm that exponentially outperforms existing classical algorithms under widely applicable assumptions. To achieve the speedup, our technique decomposes the $O(\exp(D))$-dimensional *non-sparse and full-rank* operator representing $\Sigma + \epsilon\mathbb{1}$ in the definition (3) of $q_\epsilon^*(v)$ into diagonal (i.e., sparse) operators using Fourier transform. QSVT and QFT may make our algorithm hard to simulate by classical computation, and hard to perform even on near-term quantum devices [48, 49] that cannot implement universal quantum computation due to noise. For this reason, this paper does not include numerical simulation, and we analytically prove the runtime of our algorithm. In contrast to heuristic QML algorithms for noisy quantum devices such as Ref. [48] where no proof bounds its runtime, our QML algorithm aims at applications on large scales; to achieve this aim, our proof shows the exponential advantage of our quantum algorithm over the existing classical algorithms in terms of the runtime. The wide applicability of our QML algorithm makes it a promising candidate for "killer applications" of universal quantum computers in the long run; after all, large-scale machine learning will be eventually needed in practice.

Also remarkably, since we exploit quantum computation for the sampling problem of $q_\epsilon^*(v)d\tau(v)$, our QML algorithm avoids overhead of repeating preparation of quantum states many times for estimating expectation values from the states. The classical algorithm [7, 18, 19] calculates $q_\epsilon^*(v)d\tau(v)$ by matrix inversion and then performs sampling; in contrast, our quantum algorithm never estimates classical description of $q_\epsilon^*(v)d\tau(v)$, which is represented by amplitude of a quantum state, since the overhead of such estimation would cancel out the speedup [43]. Instead, our exponential quantum speedup is achieved by performing a quantum measurement of this state to sample a feature efficiently per single preparation and measurement of the state. Our algorithm combined with stochastic gradient descent provides classical description of an estimate of $f$ to be learned, rather than a quantum state. In this way, our results discover an application of sampling problems to designing fast QML algorithms.

## 2 Setting of learning with optimized random features

### 2.1 Notation on quantum computation

In Supplementary Material, we summarize basic notions and notations of quantum computation to describe our quantum algorithms, referring to Refs. [46, 47] for more detail. An $m$-qubit quantum register is represented as a $2^m$-dimensional Hilbert space $\mathcal{H} = (\mathbb{C}^2)^{\otimes m}$. Following the conventional bra-ket notation, we represent a quantum state on the quantum register as a ket (i.e., a vector) $|\psi\rangle \in \mathcal{H}$.

### 2.2 Supervised learning with optimized random features

We introduce the supervised learning setting that we focus on in this paper, and we will formulate an approximate version of it in Sec. 2.3. Suppose that $N$ input-output pairs of examples are given by $(x_0, y_0), \ldots, (x_{N-1}, y_{N-1}) \in \mathcal{X} \times \mathcal{Y}$, where $y_n = f(x_n)$, $f : \mathcal{X} \to \mathcal{Y}$ is an unknown function to be learned, $\mathcal{X} = \mathbb{R}^D$ is the domain for $D$-dimensional input data, $\mathcal{Y} = \mathbb{R}$ is the range for output data. Each $x_n$ is an observation of an independently and identically distributed (IID) random variable on $\mathcal{X}$ equipped with a probability measure $d\rho(x) = q^{(\rho)}(x)dx$, on which we will pose a mild assumption later in Sec. 2.3. We choose a translation-invariant kernel, and such a kernel can be represented as [1]

$$k(x', x) = \int d\tau(v)\overline{\varphi(v, x')}\varphi(v, x), \ \left(\text{normalized by } k(x, x) = k(0, 0) = \int_{\mathcal{V}} d\tau(v) = 1\right), \quad (2)$$

where $\overline{\cdot}$ is complex conjugation, $\varphi : \mathcal{V} \times \mathcal{X} \to \mathbb{C}$ is a feature map $\varphi(v, x) = \mathrm{e}^{-2\pi\mathrm{i}v\cdot x}$, $\mathcal{V} = \mathbb{R}^D$ is a parameter space equipped with a probability measure $d\tau(v) = q^{(\tau)}(v)dv$, and $d\tau(v)$ is given by the Fourier transform of $k$. To specify a model of $f$, we use the RKHS $\mathcal{F}$ associated with the kernel $k$ [3]. Following Ref. [7], we assume that the norm $\|f\|_{\mathcal{F}}$ of $f$ in the RKHS is bounded, in particular, $\|f\|_{\mathcal{F}} \leqq 1$. We aim to learn an estimate $\hat{f}$ of $f$ from the $N$ given examples of data, so that the generalization error $\int d\rho(x)|\hat{f}(x) - f(x)|^2$ can be bounded to a desired learning accuracy $\epsilon > 0$.

To achieve the learning to accuracy $O(\epsilon)$ with the minimal number $M$ of features, instead of sampling from $d\tau$, Bach [7] proposes to sample features from an optimized probability density $q_\epsilon^*$ for $d\tau$

$$q_\epsilon^*(v) \propto \langle\varphi(v, \cdot)|(\Sigma + \epsilon\mathbb{1})^{-1}\varphi(v, \cdot)\rangle_{L_2(d\rho)}, \ \left(\text{normalized by } \int_{\mathcal{V}} q_\epsilon^*(v)\,d\tau(v) = 1\right), \quad (3)$$

where $\langle f \,|\, g\rangle_{L_2(d\rho)} \coloneqq \int_{\mathcal{X}} d\rho(x)\overline{f(x)}g(x)$, $\mathbb{1}$ is the identity operator, and $\Sigma : L_2(d\rho) \to L_2(d\rho)$ is the integral operator $(\Sigma f)(x') \coloneqq \int_{\mathcal{X}} d\rho(x)\,k(x', x)\,f(x)$ [50]. The function $q_\epsilon^*(v)$ is called a leverage score. Then, it suffices to sample $M$ features from $q_\epsilon^*(v)d\tau(v)$ with $M$ bounded by [7]

$$M = O(d(\epsilon)\log(d(\epsilon)/\delta)), \quad (4)$$

so as to achieve the learning to accuracy $O(\epsilon)$ with high probability greater than $1 - \delta$ for any $f$ satisfying $\|f\|_{\mathcal{F}} \leqq 1$, in formula, $\min_{\alpha_m}\{\int d\rho(x)|\hat{f}_{M,v_m,\alpha_m}(x) - f(x)|^2\} \leqq 4\epsilon$, where $\hat{f}_{M,v_m,\alpha_m}$ is the estimate (1) of $f$, and $d(\epsilon) \coloneqq \mathrm{Tr}\,\Sigma(\Sigma + \epsilon\mathbb{1})^{-1}$ is *the degree of freedom* representing effective dimension of data. In this paper, features sampled from $q_\epsilon^*(v)d\tau(v)$ up to approximation are called *optimized random features*, which achieve the learning with minimal $M$ up to a logarithmic gap [7]. In kernel methods, kernel $k$ should be chosen suitably to learn from the data given according to the distribution $d\rho$; otherwise, it is impossible for the kernel methods to achieve the learning with reasonable runtime and accuracy. In the case of random features, $f$ must have a polynomial-size description in terms of the features, i.e., $M = O(\mathrm{poly}(D, 1/\epsilon))$. To guarantee this, the degree of freedom $d(\epsilon)$ must satisfy

$$d(\epsilon) = O(\mathrm{poly}(D, 1/\epsilon)), \quad (5)$$

where $d(\epsilon)$ depends on $\Sigma$ and hence on both $d\rho$ and $k$ that is to be chosen suitably to satisfy (5).

### 2.3 Discretized representation of real number

To clarify our setting of digital quantum computation, we explain discretized representation of real number used in our quantum algorithm. We assume that the input data domain is bounded; in particular, the data distribution $d\rho(x)$ is nonzero only on a bounded domain $[0, x_{\max}]^D$ $(x_{\max} > 0)$.

Table 1: Rescaling data by $r > 1$.

| Original | Rescaled by $r > 1$ |
| --- | --- |
| $G$ of interval $[0, G]$ | $G_r = rG$ |
| Kernel $\tilde{k}(x', x)$ | $\tilde{k}_r(rx', rx) := \tilde{k}(x', x)$ |
| Input $x$ | $rx$ |
| Output $y = f(x)$ | $y = f_r(rx) := f(x)$ |
| $q^{(\rho)}(x)dx = d\rho(x)$ | $q_r^{(\rho)}(rx) := q^{(\rho)}(x)/r$ |
| $f(x)$'s LC $L^{(f)}$ | $L_r^{(f_r)} = L^{(f)}/r$ |
| $q^{(\rho)}(x)$'s LC $L^{(q^{(\rho)})}$ | $L_r^{(q_r^{(\rho)})} = L^{(q^{(\rho)})}/r^2$ |

Table 2: Discretized representation ($\tilde{x}', \tilde{x} \in \tilde{\mathcal{X}}$).

| Function / operator on $\mathcal{X}$ | Vector / operator on $\mathcal{H}^X$ |
| --- | --- |
| $f : \mathcal{X} \to \mathbb{C}$ | $\lvert f \rangle := \sum_{\tilde{x}} f(\tilde{x}) \lvert \tilde{x} \rangle$ |
| $\varphi(v, \cdot) : \mathcal{X} \to \mathbb{C}$ | $\lvert \varphi(v, \cdot) \rangle := \sum_{\tilde{x}} \varphi(v, \tilde{x}) \lvert \tilde{x} \rangle$ |
| $\tilde{k} : \mathcal{X} \times \mathcal{X} \to \mathbb{R}$ | $\mathbf{k} := \sum_{\tilde{x}', \tilde{x}} \tilde{k}(\tilde{x}', \tilde{x}) \lvert \tilde{x}' \rangle \langle \tilde{x} \rvert$ |
| $q^{(\rho)} : \mathcal{X} \to \mathbb{R}$ | $\mathbf{q}^{(\rho)} := \sum_{\tilde{x}} q^{(\rho)}(\tilde{x}) \lvert \tilde{x} \rangle \langle \tilde{x} \rvert$ |
| $\Sigma$ acting on $f : \mathcal{X} \to \mathbb{C}$ | $\boldsymbol{\Sigma} := \mathbf{k}\mathbf{q}^{(\rho)}$ |
| $\Sigma f : \mathcal{X} \to \mathbb{C}$ | $\boldsymbol{\Sigma} \lvert f \rangle$ |
| $\hat{q}^{(\rho)} : \tilde{\mathcal{X}} \to \mathbb{R}$ (Sec. 2.4) | $\hat{\mathbf{q}}^{(\rho)} := \sum_{\tilde{x}} \hat{q}^{(\rho)}(\tilde{x}) \lvert \tilde{x} \rangle \langle \tilde{x} \rvert$ |

If the kernel $k(x', x)$, such as Gaussian, decays to $0$ sufficiently fast as $x'$ and $x$ deviate from $0$, then we can approximate $k(x', x)$ using a periodic function $\tilde{k}$ with a sufficiently large period $G \gg x_{\max}$

$$k(x', x) \approx \sum_{n \in \mathbb{Z}^D} k(x', x + Gn) =: \tilde{k}(x', x), \quad \forall x', x \in [0, x_{\max}]^D. \tag{6}$$

We will use $\tilde{k}$ as a kernel in place of $k$. In computation, it is usual to represent a real number using a finite number of bits; e.g., fixed-point number representation with small precision $\Delta > 0$ uses a finite set $\{0, \Delta, 2\Delta, \dots, G - \Delta\}$ to represent a real interval $[0, G]$. Equivalently, to simplify the presentation, we use the fixed-point number representation rescaled by a parameter $r = 1/\Delta$ as shown in Table 1, so that we can use a set of integers $\mathcal{I} = \{0, 1, \dots, G_r - 1\}$ to discretize the interval. We represent the data domain $\mathcal{X} = \mathbb{R}^D$ as $\tilde{\mathcal{X}} = \mathcal{I}^D$. Discretization of the data range $\mathcal{Y}$ is unnecessary in this paper. For any real-valued point $x \in \mathcal{X}$, we write its closest grid point as $\tilde{x} \in \tilde{\mathcal{X}}$, and let $\Delta_x \subset \mathbb{R}^D$ denote a $D$-dimensional unit hypercube whose center is the closest grid point $\tilde{x}$ to $x$.

To justify this discretization, we assume that functions in the learning, such as the function $f$ to be learned and the probability density $q^{(\rho)}(x)$ of input data, are $L$-Lipschitz continuous for some Lipschitz constant (LC) $L$.[1] Then, errors caused by the discretization, i.e., $|f(x) - f(\tilde{x})|$ and $|q^{(\rho)}(x) - q^{(\rho)}(\tilde{x})|$, are negligible in the limit of small (but still nonzero) $L$, in particular, $L\sqrt{D} \to 0$. As the data dimension $D$ gets large, to reduce $L\sqrt{D}$ to a fixed error threshold, we rescale the data to a larger domain (see Table 1); in particular, we rescale $G$ representing the interval $[0, G]$ to $G_r = rG$ with $r = \Omega(L\sqrt{D})$. The rescaling in Table 1 keeps the accuracy and the model in the learning *invariant*.

We focus on asymptotic runtime analysis of our algorithm as $G_r$ gets larger, i.e., $G_r \to \infty$, which reduces the errors in the discretization. We henceforth omit the subscript $r$ and write $G_r$ as $G$ for brevity. An error analysis of discretization for finite $G$ is out of the scope of this paper; for such an analysis, we refer to established procedures in signal processing [51].

As we can represent $\tilde{\mathcal{X}}$ using $D\lceil \log_2 G \rceil$ bits, where $\lceil x \rceil$ is the least integer greater than or equal to $x$, we similarly represent $\tilde{\mathcal{X}}$ using a quantum register $\mathcal{H}^X := \text{span}\{\lvert \tilde{x} \rangle : \tilde{x} \in \tilde{\mathcal{X}}\}$ of $D\lceil \log_2 G \rceil$ qubits. This quantum register is composed of $D$ sub-registers, i.e., $\mathcal{H}^X = (\mathcal{H}_{\mathcal{I}})^{\otimes D}$, where each $\lceil \log_2 G \rceil$-qubit sub-register $\mathcal{H}_{\mathcal{I}} = (\mathbb{C}^2)^{\otimes \lceil \log_2 G \rceil}$ corresponds to $\mathcal{I}$. To represent $\tilde{x} = (\tilde{x}^{(1)}, \dots, \tilde{x}^{(D)})^{\mathrm{T}} \in \tilde{\mathcal{X}}$, we use a quantum state $\lvert \tilde{x} \rangle^X = \bigotimes_{d=1}^{D} \lvert \tilde{x}^{(d)} \rangle \in \mathcal{H}^X$, where $\lvert \tilde{x}^{(d)} \rangle \in \mathcal{H}_{\mathcal{I}}$.

We represent a function on the continuous space $\mathcal{X}$ as a vector on finite-dimensional $\mathcal{H}^X$, and an operator acting on functions on $\mathcal{X}$ as a matrix on $\mathcal{H}^X$, as shown in Table 2. Under our assumption that the rescaling makes the Lipschitz constants sufficiently small, we can make an approximation

$$\langle f | \mathbf{q}^{(\rho)} | g \rangle \approx \int_{\mathcal{X}} d\rho(x) \overline{f(x)} g(x), \quad q^{(\rho)}(x)dx = d\rho(x). \tag{7}$$

With this discretization, we can represent the optimized probability density function $q_\epsilon^*$ in (3) as

$$\tilde{q}_\epsilon^*(v) \propto \langle \varphi(v, \cdot) | \mathbf{q}^{(\rho)}(\boldsymbol{\Sigma} + \epsilon \mathbb{1})^{-1} | \varphi(v, \cdot) \rangle, \left( \text{normalized by } \int_{\mathcal{V}} \tilde{q}_\epsilon^*(v) \, d\tau(v) = 1 \right). \tag{8}$$

## 2.4 Data in discretized representation

To represent real-valued input data $x_n \in \mathcal{X}$ that is IID sampled according to the probability measure $d\rho(x)$, we use discretization. We represent $x_n$ using its closest (i.e., rounded) grid point $\tilde{x}_n \in \tilde{\mathcal{X}}$, IID sampled with probability $\int_{\Delta_{x_n}} d\rho(x)$, where $\Delta_{x_n}$ is the $D$-dimensional unit hypercube centered at $x_n$. This rounding may cause some error in learning but does not significantly ruin the performance of our QML algorithm; after all, any implementation of kernel methods by computer with bits requires rounding, and in our setting, a cluster of points that would be represented as the same grid point after the rounding are resolved by rescaling, which is equivalent to increasing precision of rounding without rescaling. Then under standard assumptions in signal processing [51] where such implementation works well, it should be straightforward to show our algorithm also works well. In the following, the $N$ given examples are $(\tilde{x}_0, y_0), \ldots, (\tilde{x}_{N-1}, y_{N-1}) \in \tilde{\mathcal{X}} \times \mathcal{Y}$, where $y_n = f(\tilde{x}_n)$.

The true probability distribution $d\rho$ of the input data is unknown in our setting, and our algorithm uses the $N$ given examples of data to approximate $d\rho(x) = q^{(\rho)}(x)dx$ up to a statistical error. For any $\tilde{x} \in \tilde{\mathcal{X}}$, we approximate the distribution $d\rho$ near $\tilde{x}$ by an empirical distribution counting the data: $\hat{q}^{(\rho)}(\tilde{x}) := {}^{n(\tilde{x})}/_N$, where $n(\tilde{x})$ denotes the number of given examples of input data that are included in the $D$-dimensional unit hypercube $\Delta_{\tilde{x}}$. We also represent $\hat{q}^{(\rho)}$ as an operator $\hat{\mathbf{q}}^{(\rho)}$ shown in Table 2. In the same way as $\mathbf{\Sigma} = \mathbf{k}\mathbf{q}^{(\rho)}$ in Table 2, an empirical integral operator is given by

$$\hat{\mathbf{\Sigma}} := \mathbf{k}\hat{\mathbf{q}}^{(\rho)}. \tag{9}$$

We aim to analyze the asymptotic runtime of our algorithm when the number $N$ of examples becomes large, as with analyzing the cases of large $G$ in the rescaling. In the limit of $N \to \infty$, statistical errors in the empirical distribution caused by the finiteness of $N$ vanish. Analysis of statistical errors for finite $N$ is out of the scope of this paper; for such an analysis, see Ref. [7].

# 3 Learning with optimized random features

We now describe our efficient quantum algorithm for sampling an optimized random feature in the setting of Sec. 2. As we show in Sec. 3.1, the novelty of our algorithm is to achieve this without assuming sparsity and low rank by means of the *perfect reconstruction* of the kernel, which decomposes the kernel by Fourier transform into a finite sum of the feature map $\varphi$ weighted over a finite set of features. In Sec. 3.2, we clarify assumptions on our quantum algorithm, bound its runtime (Theorem 1), and also show that we can achieve the learning as a whole without canceling out our quantum speedup by combining our quantum algorithm with stochastic gradient descent (Theorem 2).

Compared to existing works [20–22] on sampling random features from weighted probability distributions for acceleration, the significance of our results is that our algorithm in the limit of good approximation (as $N, G \to \infty$) is provably optimal in terms of a gap from a lower bound of the required number of random features for achieving learning [7], and yet its runtime is as fast as linear in $D$ and poly-logarithmic in $G$ (and $N$).[2] Our algorithm is constructed so as to converge to sampling from the optimized distribution (3) in Ref. [7] as $N, G \to \infty$ whereas the algorithms in Refs. [20–22] do not converge to sampling from (3) in any limit. Although the algorithms in Refs. [20–22] can achieve learning, the optimality of Refs. [20–22] is unknown in general; in contrast, Ref. [7] proves the optimality up to a logarithmic gap, and our algorithm based on Ref. [7] achieves this optimality in the limit of $N, G \to \infty$.

## 3.1 Main idea of quantum algorithm for sampling an optimized random feature

The crucial technique in our quantum algorithm is to use the perfect reconstruction of the kernel (See Proposition 1 in Supplementary Material). In the same way as representing the kernel $k$ as the expectation (2) of $\varphi(v, x) = e^{-2\pi i v \cdot x}$ over the probability distribution $d\tau = q^{(\tau)}(v)dv$, we represent our kernel $\tilde{k}$ using Shannon's sampling theorem [52] in signal processing as

$$\tilde{k}(x', x) = \sum_{\tilde{v} \in \mathbb{Z}^D} (q^{(\tau)}(\tilde{v}/G)/G^D)\overline{\varphi(\tilde{v}/G, x')}\varphi(\tilde{v}/G, x). \tag{10}$$

Table 3: Distribution $Q^{(\tau)}(v_G)$ for the Gaussian kernel (top) and the Laplacian kernel (bottom), where $v_G = (v_G^{(1)}, \ldots, v_G^{(D)})^{\mathrm{T}}$, and $\vartheta(u;q) \coloneqq 1 + 2\sum_{n=1}^{\infty} q^{n^2} \cos(2nu)$ is the theta function.

| $k(x', x)$ | $Q^{(\tau)}(v_G)$ |
|---|---|
| Gaussian kernel: $\exp(-\gamma \|x' - x\|_2^2)$ | $\prod_{d=1}^{D} \vartheta(\pi v_G^{(d)}; \exp(-\gamma))$ |
| Laplacian kernel: $\exp(-\gamma \|x' - x\|_1)$ | $\prod_{d=1}^{D} \sinh(\gamma)/(\cosh(\gamma) - \cos(2\pi v_G^{(d)}))$ |

Moreover, we show that to represent $\tilde{k}$ exactly on our *discrete* data domain $\tilde{\mathcal{X}}$, it suffices to use a *finite* set $\mathcal{V}_G$ of features and a distribution function $Q^{(\tau)}$ over the finite set $\mathcal{V}_G$

$$Q^{(\tau)}(v_G) \coloneqq \sum_{\tilde{v}' \in \mathbb{Z}^D} q^{(\tau)}(v_G + \tilde{v}'), \quad v_G \in \mathcal{V}_G \coloneqq \{0, 1/G, \ldots, 1 - 1/G\}^D, \tag{11}$$

where we give examples of $Q^{(\tau)}$ in Table 3. In particular, for all $\tilde{x}', \tilde{x} \in \tilde{\mathcal{X}}$, we show the following perfect reconstruction of our kernel $\tilde{k}(\tilde{x}', \tilde{x})$ from the function $Q^{(\tau)}$ using $D$-dimensional discrete Fourier transform $\mathbf{F}_D$ and its inverse $\mathbf{F}_D^{\dagger}$ [3]

$$\tilde{k}(\tilde{x}', \tilde{x}) = \sum_{v_G \in \mathcal{V}_G} (Q^{(\tau)}(v_G)/G^D)\overline{\varphi(v_G, \tilde{x}')}\varphi(v_G, \tilde{x}) \left( = \langle \tilde{x}' | \mathbf{F}_D^{\dagger} \mathbf{Q}^{(\tau)} \mathbf{F}_D | \tilde{x} \rangle = \langle \tilde{x}' | \mathbf{F}_D \mathbf{Q}^{(\tau)} \mathbf{F}_D^{\dagger} | \tilde{x} \rangle \right),$$

$$\tag{12}$$

where $\mathbf{Q}^{(\tau)} \coloneqq \sum_{\tilde{x} \in \tilde{\mathcal{X}}} Q^{(\tau)}(v_G) | \tilde{x} \rangle \langle \tilde{x} |$ with $v_G = \tilde{x}/G$ is a diagonal operator representing $Q^{(\tau)}$.

Thus, similarly to conventional random features using Fourier transform [1], if we sampled a sufficiently large number $M$ of features in $\mathcal{V}_G$ from the probability mass function $P^{(\tau)}(v_G) \coloneqq Q^{(\tau)}(v_G)/\left(\sum_{v'_G \in \mathcal{V}_G} Q^{(\tau)}(v'_G)\right)$ corresponding to $d\tau$, then we could combine the $M$ features with the discrete Fourier transform $\mathbf{F}_D$ to achieve the learning with the kernel $\tilde{k}(\tilde{x}', \tilde{x})$. However, $P^{(\tau)}(v_G)$ is not optimized for the data, and our quantum algorithm aims to minimize $M$ by sampling an optimized random feature. To achieve this, in place of the optimized density $\tilde{q}_{\epsilon}^*$ defined as (8) for $d\tau$ on the set $\mathcal{V}$ of real-valued features, we define an optimized probability density function $Q_{\epsilon}^*(v_G)$ for weighting the probability distribution $P^{(\tau)}(v_G)$ on the finite set $\mathcal{V}_G$ of our features as

$$Q_{\epsilon}^*(v_G) \propto \langle \varphi(v_G, \cdot) | \hat{\mathbf{q}}^{(\rho)} (\hat{\boldsymbol{\Sigma}} + \epsilon \mathbb{1})^{-1} | \varphi(v_G, \cdot) \rangle, \; \left(\text{normalized by} \sum_{v_G \in \mathcal{V}_G} Q_{\epsilon}^*(v_G) P^{(\tau)}(v_G) = 1 \right).$$

$$\tag{13}$$

To sample from optimized $Q_{\epsilon}^*(v_G) P^{(\tau)}(v_G)$, we show that we can use a quantum state on two registers $\mathcal{H}^X \otimes \mathcal{H}^{X'}$ of the same number of qubits (See Proposition 2 in Supplementary Material)

$$|\Psi\rangle^{XX'} \propto \sum_{\tilde{x} \in \tilde{\mathcal{X}}} \hat{\boldsymbol{\Sigma}}_{\epsilon}^{-\frac{1}{2}} | \tilde{x} \rangle^X \otimes \sqrt{(1/Q_{\max}^{(\tau)}) \mathbf{Q}^{(\tau)}} \mathbf{F}_D^{\dagger} \sqrt{\hat{q}^{(\rho)}(\tilde{x})} | \tilde{x} \rangle^{X'}, \tag{14}$$

where $Q_{\max}^{(\tau)} \coloneqq \max\{Q^{(\tau)}(v_G) : v_G \in \mathcal{V}_G\}$ is the maximum of $Q^{(\tau)}(v_G)$, $\hat{\boldsymbol{\Sigma}}_{\epsilon}$ is a positive semidefinite operator $\hat{\boldsymbol{\Sigma}}_{\epsilon} \coloneqq (1/Q_{\max}^{(\tau)}) \sqrt{\hat{\mathbf{q}}^{(\rho)}} \mathbf{k} \sqrt{\hat{\mathbf{q}}^{(\rho)}} + (\epsilon/Q_{\max}^{(\tau)}) \mathbb{1}$, and $f(\mathbf{A})$ for an operator $\mathbf{A}$ denotes an operator given by applying $f$ to the singular values of $\mathbf{A}$ while keeping the singular vectors, e.g., $\sqrt{\hat{\mathbf{q}}^{(\rho)}} = \sum_{\tilde{x} \in \tilde{\mathcal{X}}} \sqrt{\hat{q}^{(\rho)}(\tilde{x})} | \tilde{x} \rangle \langle \tilde{x} |$. We show that if we perform a quantum measurement of the register $\mathcal{H}^{X'}$ for the state $|\Psi\rangle^{XX'}$ in the computational basis $\{|\tilde{x}\rangle^{X'}\}$, we obtain a measurement outcome $\tilde{x}$ with probability $Q_{\epsilon}^*(\tilde{x}/G) P^{(\tau)}(\tilde{x}/G)$. Our quantum algorithm prepares $|\Psi\rangle^{XX'}$ efficiently, followed by the measurement to achieve the sampling from $Q_{\epsilon}^*(v_G) P^{(\tau)}(v_G)$, where $v_G = \tilde{x}/G$.

The difficulty in preparing the state $|\Psi\rangle^{XX'}$ arises from the fact that $|\Psi\rangle$ in (14) includes a $G^D$-dimensional operator $\hat{\boldsymbol{\Sigma}}_{\epsilon}^{-\frac{1}{2}}$, i.e. on an *exponentially large* space in $D$, and $\hat{\boldsymbol{\Sigma}}_{\epsilon}$ may *not be sparse or of low rank*. One way to use a linear operator, such as $\hat{\boldsymbol{\Sigma}}_{\epsilon}$ and $\hat{\boldsymbol{\Sigma}}_{\epsilon}^{-\frac{1}{2}}$, in quantum computation is to use the technique of block encoding [42]. In conventional ways, we can efficiently implement block encodings of *sparse or low-rank* operators [42], such as the diagonal operator $\sqrt{(1/Q_{\max}^{(\tau)}) \mathbf{Q}^{(\tau)}}$ in (14). If we had an efficient implementation of a block encoding of $\hat{\boldsymbol{\Sigma}}_{\epsilon}$, quantum singular value

transformation (QSVT) [42] would give an efficient way to implement a block encoding of $\hat{\boldsymbol{\Sigma}}_\epsilon^{-\frac{1}{2}}$ to prepare $|\Psi\rangle$. However, it has not been straightforward to discover such an efficient implementation for $\hat{\boldsymbol{\Sigma}}_\epsilon$ *without sparsity and low rank*. Recent techniques for "quantum-inspired" classical algorithms [39] are not applicable either, since the full-rank operator $\hat{\boldsymbol{\Sigma}}_\epsilon$ does not have a low-rank approximation. Remarkably, our technique does not directly use the conventional ways that require sparsity or low rank, yet implements the block encoding of $\hat{\boldsymbol{\Sigma}}_\epsilon$ efficiently.

Our significant technical contribution is to overcome the above difficulty by exploiting quantum Fourier transform (QFT) for efficient implementation of the block encoding of $\hat{\boldsymbol{\Sigma}}_\epsilon$. In our algorithm, QFTs are used for implementing the block encoding of $\hat{\boldsymbol{\Sigma}}_\epsilon$ and for applying $\mathbf{F}_D^\dagger$ in preparing $|\Psi\rangle$ in (14). The sparse and low-rank assumptions can be avoided because we explicitly decompose the (non-sparse and full-rank) operator $\hat{\boldsymbol{\Sigma}}_\epsilon$ in (14) into addition and the multiplication of diagonal (i.e., sparse) operators and QFTs. We could efficiently implement $\hat{\boldsymbol{\Sigma}}_\epsilon$ by addition and multiplication of block encodings of these diagonal operators and QFTs, but presentation of these additions and multiplications may become complicated since we have multiple block encodings to be combined. For simplicity of the presentation, we use the block encoding of the POVM operator [42] at the technical level to represent how to combine all the block encodings and QFTs as one circuit, as shown in Figs. 1 and 2 of Supplemental Material. In particular, by the perfect reconstruction (12), we decompose $\hat{\boldsymbol{\Sigma}}_\epsilon$ into diagonal operators $\sqrt{(1/Q_{\max}^{(\tau)})\mathbf{Q}^{(\tau)}}$, $\sqrt{\hat{\mathbf{q}}^{(\rho)}}$ (whose block encodings are efficiently implementable) and unitary operators $\mathbf{F}_D$, $\mathbf{F}_D^\dagger$ representing $D$-dimensional discrete Fourier transform and its inverse. The QFT provides a quantum circuit implementing $\mathbf{F}_D$ (and $\mathbf{F}_D^\dagger$) with precision $\Delta$ within time $O(D\log(G)\log(^{\log G}\!/\!_\Delta))$ [44]. We combine these implementations to obtain a quantum circuit that efficiently implements the block encoding of $\hat{\boldsymbol{\Sigma}}_\epsilon$. The QSVT of our block encoding of $\hat{\boldsymbol{\Sigma}}_\epsilon$ yields a block encoding of $\hat{\boldsymbol{\Sigma}}_\epsilon^{-\frac{1}{2}}$ with precision $\Delta$, using the block encoding of $\hat{\boldsymbol{\Sigma}}_\epsilon$ repeatedly $\widetilde{O}((Q_{\max}^{(\tau)}\!/\!\epsilon)\operatorname{polylog}(1/\Delta))$ times [42], where the factor $Q_{\max}^{(\tau)}\!/\!\epsilon$ is obtained from the condition number of $\hat{\boldsymbol{\Sigma}}_\epsilon$, and $\widetilde{O}$ may ignore poly-logarithmic factors. Using these techniques, we achieve the sampling from $Q_\epsilon^*(v_G)P^{(\tau)}(v_G)$ within a linear runtime in data dimension $D$ under the assumption that we show in the next section. See Algorithm 1 in Supplementary Material for detail.

### 3.2   Runtime analysis of learning with optimized random features

We bound the runtime of learning with optimized random features achieved by our quantum algorithm. In our runtime analysis, we use the following model of accessing given examples of data. Abstracting a device implementing random access memory (RAM) in classical computation, we assume access to the $n$th example of data via oracle functions $\mathcal{O}_{\tilde{x}}(n) = \tilde{x}_n$ and $\mathcal{O}_y(n) = y_n$ mapping $n \in \{0,\dots,N-1\}$ to the examples. Analogously to sampling $\tilde{x} \in \tilde{\mathcal{X}}$ with probability $\hat{q}(\tilde{x})$, we allow a quantum computer to use a quantum oracle (i.e., a unitary) $\mathcal{O}_\rho$ to set a quantum register $\mathcal{H}^X$ in a quantum state $\sum_{\tilde{x}\in\tilde{\mathcal{X}}}\sqrt{\hat{q}^{(\rho)}(\tilde{x})}\,|\tilde{x}\rangle$ so that we can sample $\tilde{x}$ with probability $\hat{q}(\tilde{x})$ by a measurement of this state in the computational basis $\{|\tilde{x}\rangle\}$. This input model $\mathcal{O}_\rho$ is *implementable feasibly and efficiently using techniques in Refs. [37, 53]* combined with a quantum device called quantum RAM (QRAM) [54, 55], as discussed in Supplemental Material. The oracle $\mathcal{O}_\rho$ is the only black box in our quantum algorithm; putting effort to make our algorithm explicit, we avoid any other use of QRAM. Note that the time required for accessing data is indeed a matter of computational architecture and data structure, and the focus of this paper is algorithms rather than architectures. The runtime for each query to $\mathcal{O}_{\tilde{x}}$, $\mathcal{O}_y$, and $\mathcal{O}_\rho$ is denoted by $T_{\tilde{x}}$, $T_y$, and $T_\rho$, respectively. The runtime of our algorithm does not explicitly depend on the number $N$ of given examples except that the required runtime $T_{\tilde{x}}$, $T_y$, and $T_\rho$ for accessing the data may depend on $N$, which we expect to be $O(1)$ or $O(\operatorname{polylog}(N))$.

Our algorithm can use any translation-invariant kernel $\tilde{k}$ given in the form of (12), where $Q^{(\tau)}(v_G)$ can be given by any function efficiently computable by classical computation in a short time denoted by $T_\tau = O(\operatorname{poly}(D))$, and the maximum $Q_{\max}^{(\tau)}$ of $Q^{(\tau)}(v_G)$ in (14) is also assumed to be given. We assume bounds $\tilde{k}(0,0) = \Omega(k(0,0)) = \Omega(1)$ and $Q_{\max}^{(\tau)} = O(\operatorname{poly}(D))$, which mean that the parameters of the kernel function are adjusted appropriately, so that $\tilde{k}(0,0)$ can reasonably approximate $k(0,0) = \int_{\mathcal{V}} d\tau(v) = 1$, and $Q_{\max}^{(\tau)} (= \Omega(1))$ may not be too large (e.g., not exponentially large) as $D$ gets large. Remarkably, representative choices of kernels, such as the Gaussian kernel

and the Laplacian kernel in Table 3, satisfy our assumptions in a reasonable parameter region,[4] and not only these kernels, we can use any kernel satisfying our assumptions. Our algorithm *does not impose sparsity or low rank* on $\mathbf{k}$ for the kernel and $\hat{\mathbf{q}}^{(\rho)}$ for the data distribution. Note that the requirement (5) of the upper bound of the degree of freedom $d(\epsilon)$ *does not imply low rank* of $\mathbf{k}$ and $\hat{\mathbf{q}}^{(\rho)}$ while low-rank $\mathbf{k}$ or low-rank $\hat{\mathbf{q}}^{(\rho)}$ would conversely lead to an upper bound of $d(\epsilon)$. Hence, our algorithm is widely applicable compared to existing QML algorithms shown in Sec. 1.

We prove that our quantum algorithm achieves the following runtime $T_1$. Significantly, $T_1$ is as fast as linear in $D$ whereas no existing classical algorithm achieves this sampling in sub-exponential time. Note that the precision factor $\mathrm{polylog}(1/\Delta)$ in $T_1$ of the following theorem is ignorable in practice.[5]

**Theorem 1.** *Given $D$-dimensional data discretized by $G > 0$, for any learning accuracy $\epsilon > 0$ and any sampling precision $\Delta > 0$, the runtime $T_1$ of our quantum algorithm for sampling each optimized random feature $v_G \in \mathcal{V}_G$ from a distribution $Q(v_G)P^{(\tau)}(v_G)$ close to the optimized distribution $Q_\epsilon^*(v_G)P^{(\tau)}(v_G)$ with precision $\sum_{v_G \in \mathcal{V}_G} |Q(v_G)P^{(\tau)}(v_G) - Q_\epsilon^*(v_G)P^{(\tau)}(v_G)| \leqq \Delta$ is*

$$T_1 = O(D\log(G)\log\log(G) + T_\rho + T_\tau) \times \widetilde{O}((Q_{\max}^{(\tau)}/\epsilon)\,\mathrm{polylog}(1/\Delta)).$$

Furthermore, using $M$ optimized random features $v_0, \ldots, v_{M-1}$ sampled efficiently by this quantum algorithm, we construct an algorithm achieving the learning as a whole (See Algorithm 2 in Supplementary Material), where this $M$ is to be chosen appropriately to satisfy (4). To achieve the learning, we need to obtain coefficients $\alpha_0, \ldots, \alpha_{M-1}$ of $\hat{f}_{M,v_m,\alpha_m} = \sum_{m=0}^{M-1} \alpha_m \varphi(v_m, \cdot) \approx f$ that reduce the generalization error to $O(\epsilon)$. To perform regression for obtaining $\alpha_0, \ldots, \alpha_{M-1}$, we use stochastic gradient descent (SGD) [56] (Algorithm 3 in Supplementary Material) as in the common practice of machine learning. Note that the performance of SGD with random features is extensively studied in Ref. [16], but our contribution is to clarify its *runtime* by evaluating the runtime per iteration of SGD explicitly. As discussed in Sec. 2, we aim to clarify the runtime of the learning in the large-scale limit; in particular, we assume that the number $N$ of given examples of data is sufficiently large $N > T$, where $T$ is the number of iterations in the SGD. Then, the sequence of given examples of data $(\tilde{x}_0, y_0), (\tilde{x}_1, y_1), \ldots$ provides observations of an IID random variable, and SGD converges to the minimum of the generalization error. Combining our quantum algorithm with the SGD, we achieve the following runtime $T_2$ of supervised learning with optimized random features, which is as fast as linear in $M$ and $D$, i.e., $T_2 = O(MD)$. Significantly, the required number $M$ of features for our algorithm using the optimized features is expected to be nearly *minimal*, whereas it has been computationally hard in practice to use the optimized features in classical computation.

**Theorem 2.** *(Informal) Overall runtime $T_2$ of learning with optimized random features is*

$$T_2 = O(MT_1) + O((MD + T_{\tilde{x}} + T_y) \times (1/\epsilon^2)),$$

*where $T_1$ appears in Theorem 1, the first term is the runtime of sampling $M$ optimized random features by our quantum algorithm, and the second term is the runtime of the SGD.*

# 4 Conclusion

We have constructed a quantum algorithm for sampling an *optimized random feature* within a linear time $O(D)$ in data dimension $D$, achieving an exponential speedup in $D$ compared to the existing classical algorithm [7, 18, 19] for this sampling task. Combining $M$ features sampled by this quantum algorithm with stochastic gradient descent, we can achieve supervised learning in time $O(MD)$ without canceling out the exponential speedup, where this $M$ is expected to be nearly minimal since we use the optimized random features. As for future work, it is open to prove hardness of sampling an optimized random feature for *any* possible classical algorithm under complexity-theoretical assumptions. It is also interesting to investigate whether we can reduce the runtime to $O(M\log D)$, as in Refs. [14, 15] but using the optimized random features to achieve minimal $M$. Since our quantum algorithm does not impose sparsity or low-rank assumptions, our results open a route to a widely applicable framework of kernel-based quantum machine learning with an exponential speedup.

## Broader Impact

Quantum computation has recently been attracting growing attentions owing to its potential for achieving computational speedups compared to any conventional classical computation that runs on existing computers, opening the new field of accelerating machine learning tasks via quantum computation: *quantum machine learning*. To attain a large quantum speedup, however, existing algorithms for quantum machine learning require extreme assumptions on sparsity and low rank of matrices used in the algorithms, which limit applicability of the quantum computation to machine learning tasks. In contrast, the novelty of this research is to achieve an exponential speedup in quantum machine learning without the sparsity and low-rank assumptions, broadening the applicability of quantum machine learning.

Advantageously, our quantum algorithm eliminates the computational bottleneck faced by a class of existing classical algorithms for scaling up kernel-based learning algorithms by means of random features. In particular, using this quantum algorithm, we can achieve the learning with the nearly *optimal* number of features, whereas this optimization has been hard to realize due to the bottleneck in the existing classical algorithms. A drawback of our quantum algorithm may arise from the fact that we use powerful quantum subroutines for achieving the large speedup, and these subroutines are hard to implement on existing or near-term quantum devices that cannot achieve universal quantum computation due to noise. At the same time, these subroutines make our quantum algorithm hard to simulate by classical computation, from which stems the computational advantage of our quantum algorithm over the existing classical algorithms. Thus, our results open a route to a widely applicable framework of kernel-based quantum machine learning with an exponential speedup, leading to a promising candidate of "killer applications" of universal quantum computers.

## Acknowledgments and Disclosure of Funding

This work was supported by CREST (Japan Science and Technology Agency) JPMJCR1671, Cross-ministerial Strategic Innovation Promotion Program (SIP) (Council for Science, Technologyand Innovation (CSTI)), JSPS Overseas Research Fellowships, a Cambridge-India Ramanujan scholarship from the Cambridge Trust and the SERB (Govt. of India), and JSPS KAKENHI 18K18113.

## Footnotes

[1]For any $x, x' \in \mathcal{X}$, a function $q : \mathcal{X} \to \mathbb{C}$ is $L$-Lipschitz continuous if $|q(x) - q(x')| \leqq L\|x - x'\|_2$.

[2]The runtime shown in Theorems 1 and 2 is constant time in $N$ except that classical and quantum oracles that abstract devices for accessing data may have runtime $O(1)$ or $O(\mathrm{polylog}(N))$, as discussed in Sec. 3.2.

[3]With $\mathbf{F}$ denoting a unitary operator of (one-dimensional) discrete Fourier transform, we define $\mathbf{F}_D \coloneqq \mathbf{F}^{\otimes D}$.

[4]For the kernels in Table 3, $Q^{(\tau)}$ is a product of $D$ special functions, computable in time $T_\tau = O(D)$ if each special function is computable in a constant time. It is immediate to give $Q_{\max}^{(\tau)} = Q^{(\tau)}(0)$. We have $\tilde{k}(0,0) \geqq 1 = \Omega(1)$ for these kernels. We can also fulfill $Q_{\max}^{(\tau)} = O(\mathrm{poly}(D))$ by reducing the parameter $\gamma$ of the kernels in Table 3 as $D$ increases (the reduction of $\gamma$ enlarges the class of learnable functions).

[5]E.g., inner product of $D$-dimensional real vectors is calculable in time $O(D\,\mathrm{polylog}(1/\Delta))$ with precision $\Delta$ using $O(\log(1/\Delta))$-bit fixed-point number representation, but the factor $\mathrm{polylog}(1/\Delta)$ is practically ignored.

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
