[Supplementary Material]

# Supplementary Material — Learning with Optimized Random Features: Exponential Speedup by Quantum Machine Learning without Sparsity and Low-Rank Assumptions

**Hayata Yamasaki**
The University of Tokyo, Austrian Academy of Sciences
hayata.yamasaki@gmail.com

**Sathyawageeswar Subramanian**
University of Cambridge,
University of Warwick

**Sho Sonoda**
RIKEN AIP

**Masato Koashi**
The University of Tokyo

In Supplementary Material, after summarizing basic notions of quantum computation, we provide proofs of theorems and propositions mentioned in the main text. In Sec. A, the basic notions of quantum computation are summarized. In Sec. B, the feasibility of implementing a quantum oracle that we use in our quantum algorithm is summarized. In Sec. C, we show Proposition 1 on the perfect reconstruction of the kernel, which is a crucial technique in our quantum algorithm. In Sec. D, we show Proposition 2 on a quantum state that we use in our quantum algorithm for sampling an optimized random feature. In Sec. E, we show our quantum algorithm (Algorithm 1) for sampling the optimized random feature, and prove Theorem 1 on the runtime of Algorithm 1. In Sec. F, we show the overall algorithm (Algorithm 2) for learning with the optimized random features by combining Algorithm 1 with stochastic gradient descent (Algorithm 3), and prove Theorem 2 on the runtime of Algorithm 2. Note that lemmas that we show for the runtime analysis of our quantum algorithm are presented in Sec. E, and the proofs in the other sections do not require these lemmas on quantum computation. The notations used in Supplementary Material is the same as those in the main text.

## A   Quantum computation

In this section, we summarize basic notions of quantum computation, referring to Refs. [1, 2] for more detail.

Analogously to a bit $\{0, 1\}$ in classical computation, the unit of quantum computation is a quantum bit (qubit), mathematically represented by $\mathbb{C}^2$, i.e., a 2-dimensional complex Hilbert space. A fixed orthonormal basis of a qubit $\mathbb{C}^2$ is denoted by $\{|0\rangle := \left(\begin{smallmatrix} 1 \\ 0 \end{smallmatrix}\right), |1\rangle := \left(\begin{smallmatrix} 0 \\ 1 \end{smallmatrix}\right)\}$. Similarly to a bit taking a state $b \in \{0, 1\}$, a qubit takes a quantum state $|\psi\rangle = \alpha_0 |0\rangle + \alpha_1 |1\rangle = \left(\begin{smallmatrix} \alpha_0 \\ \alpha_1 \end{smallmatrix}\right) \in \mathbb{C}^2$. While a register of $m$ bits takes values in $\{0, 1\}^m$, a quantum register of $m$ qubits is represented by the tensor-product space $\left(\mathbb{C}^2\right)^{\otimes m} \cong \mathbb{C}^{2^m}$, i.e., a $2^m$-dimensional Hilbert space. We may use $=$ rather than $\cong$ to represent isomorphism for brevity. We let $\mathcal{H}$ denote a finite-dimensional Hilbert space representing a quantum register; that is, an $m$-qubit register is $\mathcal{H} = \mathbb{C}^{2^m}$. A fixed orthonormal basis $\{|x\rangle : x \in \{0, \ldots, 2^m - 1\}\}$ labeled by $m$-bit strings, or the corresponding integers, is called the *computational basis* of $\mathcal{H}$. A state of $\mathcal{H}$ can be denoted by $|\psi\rangle = \sum_{x=0}^{2^m - 1} \alpha_x |x\rangle \in \mathcal{H}$. Any quantum state $|\psi\rangle$ requires an $L_2$ normalization condition $\||\psi\rangle\|_2 = 1$, and for any $\theta \in \mathbb{R}$, $|\psi\rangle$ is identified with $e^{i\theta} |\psi\rangle$.

In the bra-ket notation, the conjugate transpose of the column vector $|\psi\rangle$ is a row vector denoted by $\langle\psi|$, where $\langle\psi|$ and $|\psi\rangle$ may be called a bra and a ket, respectively. The inner product of $|\psi\rangle$ and $|\phi\rangle$ is denoted by $\langle\psi | \phi\rangle$, while their outer product $|\psi\rangle \langle\phi|$ is a matrix. The conjugate transpose of

an operator $\mathbf{A}$ is denoted by $\mathbf{A}^\dagger$, and the transpose of $\mathbf{A}$ with respect to the computational basis is denoted by $\mathbf{A}^{\mathrm{T}}$.

A measurement of a quantum state $|\psi\rangle$ is a sampling process that returns a randomly chosen bit string from the quantum state. An $m$-qubit state $|\psi\rangle = \sum_{x=0}^{2^m-1} \alpha_x |x\rangle$ is said to be in a superposition of the basis states $|x\rangle$s. A measurement of $|\psi\rangle$ in the computational basis $\{|x\rangle\}$ provides a random $m$-bit string $x \in \{0,1\}^m$ as outcome, with probability $p(x) = |\alpha_x|^2$. After the measurement, the state changes from $|\psi\rangle$ to $|x\rangle$ corresponding to the obtained outcome $x$, and loses the randomness in $|\psi\rangle$; that is, to iterate the same sampling as this measurement, we need to prepare $|\psi\rangle$ repeatedly for each iteration. For two registers $\mathcal{H}^A \otimes \mathcal{H}^B$ and their state $|\phi\rangle^{AB} = \sum_{x,x} \alpha_{x,x'} |x\rangle^A \otimes |x'\rangle^B \in \mathcal{H}^A \otimes \mathcal{H}^B$, a measurement of the register $\mathcal{H}^B$ for $|\phi\rangle^{AB}$ in the computational basis $\{|x'\rangle^B\}$ of $\mathcal{H}^B$ yields an outcome $x'$ with probability $p(x') = \sum_x p(x,x')$, where $p(x,x') = |\alpha_{x,x'}|^2$. The superscripts of a state or an operator represent which register the state or the operator belongs to, while we may omit the superscripts if it is clear from the context.

A quantum algorithm starts by initializing $m$ qubits in a fixed state $|0\rangle^{\otimes m}$, which we may write as $|0\rangle$ if $m$ is clear from the context. Then, we apply a $2^m$-dimensional unitary operator $\mathbf{U}$ to $|0\rangle^{\otimes m}$, to prepare a state $\mathbf{U}|0\rangle^{\otimes m}$. Finally, a measurement of $\mathbf{U}|0\rangle^{\otimes n}$ is performed to sample an $m$-bit string from a probability distribution given by $\mathbf{U}|0\rangle^{\otimes m}$. Analogously to classical logic-gate circuits, $\mathbf{U}$ is represented by a quantum circuit composed of sequential applications of unitaries acting at most two qubits at a time. Each of these unitaries is called an elementary quantum gate. The runtime of a quantum algorithm represented by a quantum circuit is determined by the number of applications of elementary quantum gates in the circuit.

With techniques shown in Refs. [3–5], non-unitary operators can also be used in quantum computation. In particular, to apply a non-unitary operator $\mathbf{A}$ in quantum computation, we use the technique of *block encoding* [5], as summarized in the following. A block encoding of $\mathbf{A}$ is a unitary operator $\mathbf{U} = \left(\begin{smallmatrix} \mathbf{A} & \cdot \\ \cdot & \cdot \end{smallmatrix}\right)$ that encodes $\mathbf{A}$ in its left-top (or $|0\rangle\langle 0|$) subspace (up to numerical precision). Note that we have

$$\mathbf{U} = \left(\begin{smallmatrix} \mathbf{A} & \mathbf{B} \\ \mathbf{C} & \mathbf{D} \end{smallmatrix}\right) = |0\rangle\langle 0| \otimes \mathbf{A} + |0\rangle\langle 1| \otimes \mathbf{B} + |1\rangle\langle 0| \otimes \mathbf{C} + |1\rangle\langle 1| \otimes \mathbf{D}, \tag{1}$$

if $\mathbf{A}$, $\mathbf{B}$, $\mathbf{C}$, and $\mathbf{D}$ are on the Hilbert space of the same dimension. Consider a state $|0\rangle \otimes |\psi\rangle = \left(\begin{smallmatrix} |\psi\rangle \\ \mathbf{0} \end{smallmatrix}\right)$ in the top-left (or $|0\rangle\langle 0|$) subspace of $\mathbf{U}$, where $\mathbf{0}$ is a zero column vector, and $|0\rangle \in \mathbb{C}^d$ for some $d$. Applying $\mathbf{U}$ to the state $|0\rangle \otimes |\psi\rangle$, we would obtain

$$\mathbf{U}(|0\rangle \otimes |\psi\rangle) = \sqrt{p}\,|0\rangle \otimes \frac{\mathbf{A}|\psi\rangle}{\|\mathbf{A}|\psi\rangle\|_2} + \sqrt{1-p}\,|\perp\rangle, \tag{2}$$

where $p = \|\mathbf{A}|\psi\rangle\|_2^2$, and $|\perp\rangle$ is a state of no interest satisfying $(|0\rangle\langle 0| \otimes \mathbb{1})|\perp\rangle$. Then, we can prepare the state to which $\mathbf{A}$ is applied, i.e.,

$$\frac{\mathbf{A}|\psi\rangle}{\|\mathbf{A}|\psi\rangle\|_2} \tag{3}$$

using this process for preparing $\mathbf{U}(|0\rangle \otimes |\psi\rangle)$ and its inverse process repeatedly $O(\frac{1}{\sqrt{p}})$ times, by means of amplitude amplification [6]. Note that given a quantum circuit, its inverse can be implemented by replacing each gate in the circuit with its inverse gate; that is, the circuit and its inverse circuit have the same runtime since they are composed of the same number of gates. In Sec. E, we will use the following more precise definition of block encoding to take the precision $\Delta$ into account. For any operator $\mathbf{A}$ on $s$ qubits, i.e., on $\mathbb{C}^{2^s}$, a unitary operator $\mathbf{U}$ on $(s+a)$ qubits, i.e., on $\mathbb{C}^{2^{s+a}}$, is called an $(\alpha, a, \Delta)$-block encoding of $\mathbf{A}$ if it holds that

$$\left\| \mathbf{A} - \alpha \left(\mathbb{1} \otimes \langle 0|^{\otimes a}\right) \mathbf{U} \left(\mathbb{1} \otimes |0\rangle^{\otimes a}\right) \right\|_\infty \leqq \Delta, \tag{4}$$

where $\|\cdot\|_\infty$ is the operator norm. Note that since any unitary operator $\mathbf{U}$ satisfies $\|\mathbf{U}\|_\infty \leqq 1$, it is necessary that $\|\mathbf{A}\|_\infty \leqq \alpha + \Delta$.

## B   Feasibility of implementing quantum oracle

In this section, we summarize the feasibility of implementing a quantum oracle that we use in our quantum algorithm.

The quantum oracles are mathematically represented by unitary operators. As shown in the main text, to access given examples of data in our quantum algorithm, we use a quantum oracle $\mathcal{O}_\rho$ acting as

$$\mathcal{O}_\rho(|0\rangle) = \sum_{\tilde{x} \in \tilde{\mathcal{X}}} \sqrt{\hat{q}^{(\rho)}(\tilde{x})} \, |\tilde{x}\rangle = \sqrt{\hat{\mathbf{q}}^{(\rho)}} \sum_{\tilde{x} \in \tilde{\mathcal{X}}} |\tilde{x}\rangle \,, \tag{5}$$

where we write

$$\hat{\mathbf{q}}^{(\rho)} = \sum_{\tilde{x} \in \tilde{\mathcal{X}}} \hat{q}^{(\rho)}(\tilde{x}) \, |\tilde{x}\rangle \langle \tilde{x}| \,. \tag{6}$$

We can efficiently implement the quantum oracle $\mathcal{O}_\rho$ with an acceptable preprocessing overhead using the $N$ given examples of input data $\tilde{x}_0, \ldots, \tilde{x}_{N-1}$. From these examples, we can prepare a data structure proposed in Ref. [7] in $O(N(D \log G)^2)$ time using $O(N(D \log G)^2)$ bits of memory, while collecting and storing the $N$ data points requires at least $\Theta(ND \log G)$ time and $\Theta(ND \log G)$ bits of memory. Note that this data structure is also used in "quantum-inspired" classical algorithms [8–10]. Then, we can implement $\mathcal{O}_\rho$ by a quantum circuit combined with a quantum random access memory (QRAM) [11, 12], which can load data from this data structure into qubits in quantum superposition (i.e. linear combinations of quantum states). With $T_Q$ denoting runtime of this QRAM per query, it is known that this implementation of $\mathcal{O}_\rho$ with precision $\Delta$ has runtime

$$T_\rho = O(D \log(G) \operatorname{polylog}(1/\Delta) \times T_Q) \tag{7}$$

per query [7, 13]. The runtime $T_Q$ of this QRAM may scale poly-logarithmically in $N$ depending on how we implement the QRAM, but such an implementation suffices to meet our expectation in the main text that $T_\rho$ should be $O(1)$ or $O(\operatorname{polylog}(N))$ as $N$ increases. Note that the inverse $\mathcal{O}_\rho^\dagger$ of $\mathcal{O}_\rho$ has the same runtime $T_\rho$ since $\mathcal{O}_\rho^\dagger$ can be implemented by replacing each quantum gate in the circuit for $\mathcal{O}_\rho$ with its inverse.

Thus, if both the quantum computer and the QRAM are available, we can implement $\mathcal{O}_\rho$ feasibly and efficiently. Similarly to the quantum computer assumed to be available in this paper, QRAM is actively under development towards its physical realization; e.g., see Ref. [14] on recent progress towards realizing QRAM. The use of QRAM is a common assumption in quantum machine learning (QML) especially to deal with a large amount of data; however, even with QRAM, achieving quantum speedup is nontrivial. Note that we do not include the time for collecting the data or preparing the above data structure in runtime of our learning algorithm, but even if we took them into account, an exponential speedup from $O(\exp(D))$ to $O(\operatorname{poly}(D))$ would not be canceled out. Since we exploit $\mathcal{O}_\rho$ for constructing a widely applicable QML framework achieving the exponential speedup without sparsity and low-rank assumptions, our results motivate further technological development towards realizing the QRAM as well as the quantum computer.

## C  Perfect reconstruction of kernel

In this section, we show the following perfect reconstruction of the kernel that we use in our quantum algorithm.

**Proposition 1** (Perfect reconstruction of kernel). *Given any periodic translation-invariant kernel $\tilde{k}$, we exactly have for each $\tilde{x}', \tilde{x} \in \tilde{\mathcal{X}}$*

$$\tilde{k}(\tilde{x}', \tilde{x}) = \sum_{v_G \in \mathcal{V}_G} \frac{Q^{(\tau)}(v_G)}{G^D} \overline{\varphi(v_G, \tilde{x}')} \varphi(v_G, \tilde{x})$$

$$= \langle \tilde{x}' | \, \mathbf{F}_D^\dagger \mathbf{Q}^{(\tau)} \mathbf{F}_D \, | \tilde{x} \rangle = \langle \tilde{x}' | \, \mathbf{F}_D \mathbf{Q}^{(\tau)} \mathbf{F}_D^\dagger \, | \tilde{x} \rangle \,.$$

*Proof.* To show the perfect reconstruction of the kernel $\tilde{k}$, we crucially use the assumption given in the main text that the data domain is finite due to the discretized representation

$$\tilde{\mathcal{X}} = \{0, 1, \ldots, G-1\}^D. \tag{8}$$

As summarized in the main text, recall that we approximate a translation-invariant (but not necessarily periodic) kernel $k(x', x)$ by

$$\tilde{k}(x', x) = \sum_{n \in \mathbb{Z}^D} k(x', x + Gn). \tag{9}$$

To represent the translation-invariant kernel functions, we may write

$$k_{\mathrm{TI}}\left(x'-x\right) := k\left(x',x\right), \tag{10}$$

$$\tilde{k}_{\mathrm{TI}}\left(x'-x\right) := \tilde{k}\left(x',x\right). \tag{11}$$

The function $\tilde{k}$ is periodic by definition; in particular, we have for any $n' \in \mathbb{Z}^D$

$$\tilde{k}\left(x',x\right) = \tilde{k}\left(x'+Gn',x\right) = \tilde{k}\left(x',x+Gn'\right) = \tilde{k}_{\mathrm{TI}}(x'-x+Gn'). \tag{12}$$

Recall that the translation-invariant kernel $k : \mathcal{X} \times \mathcal{X} \to \mathbb{R}$ can be written as

$$k\left(x',x\right) = \int_{\mathcal{V}} d\tau\left(v\right) \overline{\varphi\left(v,x'\right)} \varphi\left(v,x\right), \tag{13}$$

where $\varphi(v,x) := \mathrm{e}^{-2\pi \mathrm{i}v\cdot x}$, and $d\tau$ is given by the Fourier transform of the kernel, in particular, [15]

$$d\tau(v) = q^{(\tau)}(v)dv = \left[\int_{\mathcal{X}} dx\, \mathrm{e}^{-2\pi \mathrm{i}v\cdot x} k_{\mathrm{TI}}\left(x\right)\right] dv. \tag{14}$$

Similarly to (14), our proof will expand $\tilde{k}$ using the Fourier transform.

To expand $\tilde{k}$, we first consider the case of $D = 1$, and will later consider $D \geqq 1$ in general. In the case of $D = 1$, Shannon's sampling theorem [16] in signal processing [17] shows that we can perfectly reconstruct the kernel function $\tilde{k}_{\mathrm{TI}}$ on a *continuous* domain $\left[-\frac{G}{2}, \frac{G}{2}\right]$ from *discrete* frequencies of its Fourier transform. In the one-dimensional case, the Fourier transform of $\tilde{k}_{\mathrm{TI}}$ on $\left[-\frac{G}{2}, \frac{G}{2}\right]$ is

$$\int_{-\frac{G}{2}}^{\frac{G}{2}} dx\, \tilde{k}_{\mathrm{TI}}(x)\mathrm{e}^{-2\pi \mathrm{i}vx} = \int_{-\infty}^{\infty} dx\, k_{\mathrm{TI}}(x)\mathrm{e}^{-2\pi \mathrm{i}vx} = q^{(\tau)}(v). \tag{15}$$

Then, for any $x \in \left[-\frac{G}{2}, \frac{G}{2}\right]$, using the discrete frequencies $\tilde{v} \in \mathbb{Z}$ for $q^{(\tau)}(\tilde{v})$, we exactly obtain from the sampling theorem

$$\tilde{k}_{\mathrm{TI}}\left(x\right) = \frac{1}{G} \sum_{\tilde{v}=-\infty}^{\infty} q^{(\tau)}\left(\frac{\tilde{v}}{G}\right) \mathrm{e}^{2\pi \mathrm{i}(\frac{\tilde{v}}{G})x} = \frac{1}{G} \sum_{\tilde{v}=-\infty}^{\infty} q^{(\tau)}\left(\frac{\tilde{v}}{G}\right) \mathrm{e}^{\frac{2\pi \mathrm{i}\tilde{v}x}{G}}. \tag{16}$$

Due to the periodicity (12) of $\tilde{k}_{\mathrm{TI}}$, (16) indeed holds for any $x \in \mathbb{R}$. In the same way, for any $D \geqq 1$, we have for any $x \in \mathbb{R}^D$

$$\tilde{k}_{\mathrm{TI}}\left(x\right) = \frac{1}{G^D} \sum_{\tilde{v}\in\mathbb{Z}^D} q^{(\tau)}\left(\frac{\tilde{v}}{G}\right) \mathrm{e}^{\frac{2\pi \mathrm{i}\tilde{v}\cdot x}{G}}. \tag{17}$$

In addition, since $\tilde{\mathcal{X}}$ is a *discrete* domain spaced at intervals 1, we can achieve the perfect reconstruction of the kernel $\tilde{k}_{\mathrm{TI}}$ on $\tilde{\mathcal{X}}$ by the $D$-dimensional discrete Fourier transform of $\tilde{k}_{\mathrm{TI}}$, using a *finite* set of discrete frequencies for $q^{(\tau)}$. In particular, for each $\tilde{v} \in \tilde{\mathcal{X}}$, the discrete Fourier transform of $\tilde{k}_{\mathrm{TI}}$ yields

$$\frac{1}{\sqrt{G^D}} \sum_{\tilde{x}\in\tilde{\mathcal{X}}} \tilde{k}_{\mathrm{TI}}\left(\tilde{x}\right) \mathrm{e}^{\frac{-2\pi \mathrm{i}\tilde{v}\cdot\tilde{x}}{G}} = \frac{1}{\sqrt{G^D}} \sum_{\tilde{x}\in\tilde{\mathcal{X}}} \left(\frac{1}{G^D} \sum_{\tilde{v}''\in\mathbb{Z}^D} q^{(\tau)}\left(\frac{\tilde{v}''}{G}\right) \mathrm{e}^{\frac{2\pi \mathrm{i}\tilde{v}''\cdot\tilde{x}}{G}}\right) \mathrm{e}^{\frac{-2\pi \mathrm{i}\tilde{v}\cdot\tilde{x}}{G}}$$

$$= \frac{1}{\sqrt{G^D}} \sum_{\tilde{v}'\in\mathbb{Z}^D} q^{(\tau)}\left(\frac{\tilde{v}}{G} + \tilde{v}'\right), \tag{18}$$

where the sum over $\tilde{x}$ in the first line is nonzero if $\tilde{v}'' = \tilde{v} + G\tilde{v}'$ for any $\tilde{v}' \in \mathbb{Z}^D$. Thus for the perfect reconstruction of the kernel $\tilde{k}$ on this domain $\tilde{\mathcal{X}}$, it suffices to use feature points $v_G = \frac{\tilde{v}}{G}$ for each $\tilde{v} \in \tilde{\mathcal{X}}$, which yields a finite set $\mathcal{V}_G$ of features

$$v_G = \begin{pmatrix} v_G^{(1)} \\ \vdots \\ v_G^{(D)} \end{pmatrix} \in \mathcal{V}_G := \left\{0, \frac{1}{G}, \ldots, 1 - \frac{1}{G}\right\}^D. \tag{19}$$

We use the one-to-one correspondence between $v_G \in \mathcal{V}_G$ and $\tilde{x} \in \tilde{\mathcal{X}}$ satisfying

$$v_G = \frac{\tilde{x}}{G},\qquad(20)$$

which we may also write using $\tilde{v} = \tilde{x}$ as

$$v_G = \frac{\tilde{v}}{G}.\qquad(21)$$

In the same way as the main text, we let $Q^{(\tau)} : \mathcal{V}_G \to \mathbb{R}$ denote the function in (18)

$$Q^{(\tau)}(v_G) := \sum_{\tilde{v}' \in \mathbb{Z}^D} q^{(\tau)}(v_G + \tilde{v}').\qquad(22)$$

Therefore, from the $D$-dimensional discrete Fourier transform of (18), we obtain the perfect reconstruction of the kernel $\tilde{k}_{\text{TI}}$ on the domain $\tilde{\mathcal{X}}$ using the feature points in $\mathcal{V}_G$ and the function $Q^{(\tau)}$ as

$$\tilde{k}(\tilde{x}', \tilde{x}) = \tilde{k}_{\text{TI}}(\tilde{x}' - \tilde{x})$$

$$= \frac{1}{\sqrt{G^D}} \sum_{\tilde{v} \in \tilde{\mathcal{X}}} \left( \frac{1}{\sqrt{G^D}} \sum_{\tilde{v}' \in \mathbb{Z}^D} q^{(\tau)}\left( \frac{\tilde{v}}{G} + \tilde{v}' \right) \right) \mathrm{e}^{\frac{2\pi \mathrm{i} \tilde{v} \cdot (\tilde{x}' - \tilde{x})}{G}}$$

$$= \sum_{v_G \in \mathcal{V}_G} \frac{Q^{(\tau)}(v_G)}{G^D} \overline{\varphi(v_G, \tilde{x}')} \varphi(v_G, \tilde{x}), \quad \forall \tilde{x}', \tilde{x} \in \tilde{\mathcal{X}},\qquad(23)$$

which shows the first equality in Proposition 1. Note that this equality also leads to a lower bound of $Q^{(\tau)}_{\max}$, that is, the maximum of $Q^{(\tau)}(v_G)$, as shown in Remark 1 after this proof.

To show the second equality in Proposition 1, recall that we write a diagonal operator corresponding to $Q^{(\tau)}(v_G)$ as

$$\mathbf{Q}^{(\tau)} := \sum_{\tilde{v} \in \tilde{\mathcal{X}}} Q^{(\tau)}\left( \frac{\tilde{v}}{G} \right) |\tilde{v}\rangle \langle \tilde{v}|.\qquad(24)$$

Note that we write $|\tilde{v}\rangle = |\tilde{x}\rangle$ for $\tilde{v} = \tilde{x} \in \tilde{\mathcal{X}}$ for clarity of the presentation. In addition, let $\mathbf{F}$ denote a unitary operator representing (one-dimensional) discrete Fourier transform

$$\mathbf{F} := \sum_{\tilde{x}=0}^{G-1} \left( \frac{1}{\sqrt{G}} \sum_{\tilde{v}=0}^{G-1} \mathrm{e}^{-\frac{2\pi \mathrm{i} \tilde{v} \tilde{x}}{G}} |\tilde{v}\rangle \right) \langle \tilde{x}|,\qquad(25)$$

and $\mathbf{F}_D$ denote a unitary operator representing $D$-dimensional discrete Fourier transform

$$\mathbf{F}_D := \mathbf{F}^{\otimes D} = \sum_{\tilde{x} \in \tilde{\mathcal{X}}} \left( \frac{1}{\sqrt{G^D}} \sum_{\tilde{v} \in \tilde{\mathcal{X}}} \mathrm{e}^{-\frac{2\pi \mathrm{i} \tilde{v} \cdot \tilde{x}}{G}} |\tilde{v}\rangle \right) \langle \tilde{x}|.\qquad(26)$$

The feature map can be written in terms of $\mathbf{F}_D$ as

$$\varphi(v_G, \tilde{x}) = \mathrm{e}^{-2\pi \mathrm{i} v_G \cdot \tilde{x}} = \sqrt{G^D} \langle \tilde{v} | \mathbf{F}_D | \tilde{x} \rangle = \sqrt{G^D} \langle \tilde{x} | \mathbf{F}_D | \tilde{v} \rangle,\qquad(27)$$

where $v_G = \frac{\tilde{v}}{G}$, and the last equality follows from the invariance of $\mathbf{F}_D$ under the transpose with respect to the computational basis. From (23), (26), and (27), by linear algebraic calculation, we obtain the conclusion for any $\tilde{x}', \tilde{x} \in \tilde{\mathcal{X}}$

$$\tilde{k}(\tilde{x}', \tilde{x}) = \left\langle \tilde{x}' \left| \mathbf{F}_D^\dagger \mathbf{Q}^{(\tau)} \mathbf{F}_D \right| \tilde{x} \right\rangle = \left\langle \tilde{x}' \left| \mathbf{F}_D \mathbf{Q}^{(\tau)} \mathbf{F}_D^\dagger \right| \tilde{x} \right\rangle,\qquad(28)$$

where the last equality follows from the fact that the kernel function $\tilde{k}$ is symmetric and real, i.e., $\tilde{k}(x', x) = \tilde{k}(x, x')$ and $\overline{\tilde{k}(x', x)} = \tilde{k}(x', x)$. $\qquad\square$

*Remark* 1 (A lower bound of $Q_{\max}^{(\tau)}$). Equality (23) has the following implication on a lower bound of the maximum of $Q^{(\tau)}(v_G)$

$$Q_{\max}^{(\tau)} = \max\left\{ Q^{(\tau)}(v_G) : v_G \in \mathcal{V}_G \right\}. \tag{29}$$

Recall that we let $P^{(\tau)}$ denote a probability mass function on $\mathcal{V}_G$ proportional to $Q^{(\tau)}$

$$P^{(\tau)}(v_G) := \frac{Q^{(\tau)}(v_G)}{\sum_{v'_G \in \mathcal{V}_G} Q^{(\tau)}(v'_G)}, \tag{30}$$

which by definition satisfies the normalization condition

$$\sum_{v_G \in \mathcal{V}_G} P^{(\tau)}(v_G) = 1. \tag{31}$$

We obtain from (23)

$$\tilde{k}(0,0) = \sum_{v_G \in \mathcal{V}_G} \frac{Q^{(\tau)}(v_G)}{G^D}, \tag{32}$$

and hence, we can regard $\tilde{k}(0,0)$ as a normalization factor in

$$P^{(\tau)}(v_G) = \frac{1}{\tilde{k}(0,0)} \frac{Q^{(\tau)}(v_G)}{G^D}. \tag{33}$$

The normalization of $P^{(\tau)}$ yields a lower bound of $Q_{\max}^{(\tau)}$

$$Q_{\max}^{(\tau)} = G^D \times \frac{Q_{\max}^{(\tau)}}{G^D} \geqq \sum_{v_G \in \mathcal{V}_G} \frac{Q^{(\tau)}(v_G)}{G^D} = \tilde{k}(0,0) \sum_{v_G \in \mathcal{V}_G} P^{(\tau)}(v_G) = \tilde{k}(0,0) = \Omega(1), \tag{34}$$

where we use the assumption $\tilde{k}(0,0) = \Omega(k(0,0)) = \Omega(1)$.

## D   Quantum state for sampling an optimized random feature

In this section, we show a quantum state that we use in our quantum algorithm for sampling an optimized random feature. In particular, as shown in the main text, recall a quantum state on two quantum registers $\mathcal{H}^X \otimes \mathcal{H}^{X'}$

$$|\Psi\rangle^{XX'} \propto \sum_{\tilde{x} \in \tilde{\mathcal{X}}} \hat{\Sigma}_\epsilon^{-\frac{1}{2}} |\tilde{x}\rangle^X \otimes \sqrt{\frac{1}{Q_{\max}^{(\tau)}} \mathbf{Q}^{(\tau)}} \mathbf{F}_D^\dagger \sqrt{\hat{q}^{(\rho)}(\tilde{x})} |\tilde{x}\rangle^{X'}, \tag{35}$$

where $X$ and $X'$ have the same number of qubits. Then, we show the following proposition.

**Proposition 2** (Quantum state for sampling an optimized random feature)**.** *If we perform a measurement of the quantum register $X'$ on the state $|\Psi\rangle^{XX'}$ defined as (35) in the computational basis $\{|\tilde{x}\rangle^{X'} : \tilde{x} \in \tilde{\mathcal{X}}\}$, then we obtain a measurement outcome $\tilde{x}$ with probability $Q_\epsilon^*\left(\frac{\tilde{x}}{G}\right) P^{(\tau)}\left(\frac{\tilde{x}}{G}\right)$.*

*Proof.* The proof is given by linear algebraic calculation. Note that the normalization $\left\||\Psi\rangle^{XX'}\right\|_2 = 1$ of a quantum state always yields the normalization $\sum_{\tilde{x}' \in \tilde{\mathcal{X}}} p(\tilde{x}') = 1$ of a probability distribution obtained from the measurement of $\mathcal{H}^{X'}$ in the computational basis $\{|\tilde{x}'\rangle^{X'}\}$, and hence, we may omit the normalization constant in the following calculation for simplicity of the presentation.

Recall the definition of the optimized probability distribution $Q_\epsilon^*(v_G) P^{(\tau)}(v_G)$

$$Q_\epsilon^*(v_G) P^{(\tau)}(v_G) = \frac{\left\langle \varphi(v_G, \cdot) \left| \hat{\mathbf{q}}^{(\rho)}\left(\hat{\Sigma} + \epsilon \mathbb{1}\right)^{-1} \right| \varphi(v_G, \cdot) \right\rangle Q^{(\tau)}(v_G)}{\sum_{v'_G \in \mathcal{V}_G} \left\langle \varphi(v'_G, \cdot) \left| \hat{\mathbf{q}}^{(\rho)}\left(\hat{\Sigma} + \epsilon \mathbb{1}\right)^{-1} \right| \varphi(v'_G, \cdot) \right\rangle Q^{(\tau)}(v'_G)}, \tag{36}$$

where we write

$$\hat{\Sigma} = \mathbf{k}\hat{\mathbf{q}}^{(\rho)}, \tag{37}$$

$$\mathbf{k} = \sum_{\tilde{x}',\tilde{x}\in\tilde{\mathcal{X}}} \tilde{k}\left(\tilde{x}',\tilde{x}\right)|\tilde{x}'\rangle\langle\tilde{x}|. \tag{38}$$

For $v_G = \frac{\tilde{x}}{G}$, it follows from (27) that

$$|\varphi\left(v_G,\cdot\right)\rangle = \sqrt{G^D}\mathbf{F}_D|\tilde{x}\rangle. \tag{39}$$

Then, we have

$$(36) = \frac{\left\langle\varphi\left(v_G,\cdot\right)\left|\hat{\mathbf{q}}^{(\rho)}\left(\hat{\Sigma}+\epsilon\mathbb{1}\right)^{-1}\right|\varphi\left(v_G,\cdot\right)\right\rangle}{\sum_{v'_G\in\mathcal{V}_G}\frac{Q^{(\tau)}\left(v'_G\right)}{G^D}\left\langle\varphi\left(v'_G,\cdot\right)\left|\hat{\mathbf{q}}^{(\rho)}\left(\hat{\Sigma}+\epsilon\mathbb{1}\right)^{-1}\right|\varphi\left(v'_G,\cdot\right)\right\rangle}\frac{Q^{(\tau)}\left(v_G\right)}{G^D}$$

$$= \frac{\left\langle\tilde{x}\left|\mathbf{F}_D^\dagger\hat{\mathbf{q}}^{(\rho)}\left(\hat{\Sigma}+\epsilon\mathbb{1}\right)^{-1}\mathbf{F}_D\right|\tilde{x}\right\rangle}{\sum_{\tilde{x}'\in\tilde{\mathcal{X}}}Q^{(\tau)}\left(\frac{\tilde{x}'}{G}\right)\left\langle\tilde{x}'\left|\mathbf{F}_D^\dagger\hat{\mathbf{q}}^{(\rho)}\left(\hat{\Sigma}+\epsilon\mathbb{1}\right)^{-1}\mathbf{F}_D\right|\tilde{x}'\right\rangle}Q^{(\tau)}\left(\frac{\tilde{x}}{G}\right). \tag{40}$$

Then, using (24), we obtain

$$(40) = \frac{\left\langle\tilde{x}\left|\sqrt{\mathbf{Q}^{(\tau)}}\mathbf{F}_D^\dagger\hat{\mathbf{q}}^{(\rho)}\left(\hat{\Sigma}+\epsilon\mathbb{1}\right)^{-1}\mathbf{F}_D\sqrt{\mathbf{Q}^{(\tau)}}\right|\tilde{x}\right\rangle}{\sum_{\tilde{x}'\in\tilde{\mathcal{X}}}\left\langle\tilde{x}'\left|\sqrt{\mathbf{Q}^{(\tau)}}\mathbf{F}_D^\dagger\hat{\mathbf{q}}^{(\rho)}\left(\hat{\Sigma}+\epsilon\mathbb{1}\right)^{-1}\mathbf{F}_D\sqrt{\mathbf{Q}^{(\tau)}}\right|\tilde{x}'\right\rangle}. \tag{41}$$

Therefore, it holds that

$$Q_\epsilon^*\left(\frac{\tilde{x}}{G}\right)P^{(\tau)}\left(\frac{\tilde{x}}{G}\right) \propto \left\langle\tilde{x}\left|\sqrt{\mathbf{Q}^{(\tau)}}\mathbf{F}_D^\dagger\hat{\mathbf{q}}^{(\rho)}\left(\hat{\Sigma}+\epsilon\mathbb{1}\right)^{-1}\mathbf{F}_D\sqrt{\mathbf{Q}^{(\tau)}}\right|\tilde{x}\right\rangle$$

$$= \left\langle\tilde{x}\left|\sqrt{\frac{1}{Q_{\max}^{(\tau)}}\mathbf{Q}^{(\tau)}}\mathbf{F}_D^\dagger\hat{\mathbf{q}}^{(\rho)}\left(\frac{1}{Q_{\max}^{(\tau)}}\hat{\Sigma}+\frac{\epsilon}{Q_{\max}^{(\tau)}}\mathbb{1}\right)^{-1}\mathbf{F}_D\sqrt{\frac{1}{Q_{\max}^{(\tau)}}\mathbf{Q}^{(\tau)}}\right|\tilde{x}\right\rangle. \tag{42}$$

To simplify the form of (42), define a positive semidefinite operator on the support of $\hat{\mathbf{q}}^{(\rho)}$

$$\hat{\Sigma}_\epsilon^{(\rho)} := \sqrt{\hat{\mathbf{q}}^{(\rho)}}\left(\frac{1}{Q_{\max}^{(\tau)}}\hat{\Sigma}+\frac{\epsilon}{Q_{\max}^{(\tau)}}\mathbb{1}\right)\left(\hat{\mathbf{q}}^{(\rho)}\right)^{-\frac{1}{2}}$$

$$= \frac{1}{Q_{\max}^{(\tau)}}\sqrt{\hat{\mathbf{q}}^{(\rho)}}\mathbf{k}\sqrt{\hat{\mathbf{q}}^{(\rho)}}+\frac{\epsilon}{Q_{\max}^{(\tau)}}\Pi^{(\rho)}, \tag{43}$$

where we use $\hat{\Sigma} = \mathbf{k}\hat{\mathbf{q}}^{(\rho)}$, and $\Pi^{(\rho)}$ is a projector onto the support of $\hat{\mathbf{q}}^{(\rho)}$. In case $\hat{\mathbf{q}}^{(\rho)}$ does not have full rank, $\left(\hat{\mathbf{q}}^{(\rho)}\right)^{-\frac{1}{2}}$ denotes $\sqrt{\left(\hat{\mathbf{q}}^{(\rho)}\right)^{-1}}$, where $\left(\hat{\mathbf{q}}^{(\rho)}\right)^{-1}$ in this case is the Moore-Penrose pseudoinverse of $\hat{\mathbf{q}}^{(\rho)}$. We have by definition

$$\left(\hat{\Sigma}_\epsilon^{(\rho)}\right)^{-1} = \sqrt{\hat{\mathbf{q}}^{(\rho)}}\left(\frac{1}{Q_{\max}^{(\tau)}}\hat{\Sigma}+\frac{\epsilon}{Q_{\max}^{(\tau)}}\mathbb{1}\right)^{-1}\left(\hat{\mathbf{q}}^{(\rho)}\right)^{-\frac{1}{2}}. \tag{44}$$

Correspondingly, in the same way as the main text, we let $\hat{\Sigma}_\epsilon$ denote a positive definite operator that has the full support on $\mathcal{H}^X$, and coincides with $\hat{\Sigma}_\epsilon^{(\rho)}$ if projected on the support of $\hat{\mathbf{q}}^{(\rho)}$

$$\hat{\Sigma}_\epsilon := \frac{1}{Q_{\max}^{(\tau)}}\sqrt{\hat{\mathbf{q}}^{(\rho)}}\mathbf{k}\sqrt{\hat{\mathbf{q}}^{(\rho)}}+\frac{\epsilon}{Q_{\max}^{(\tau)}}\mathbb{1}. \tag{45}$$

Note that since $\hat{\mathbf{q}}^{(\rho)}$ is diagonal and $\mathbf{k}$ is symmetric, we have

$$\hat{\mathbf{\Sigma}}_\epsilon = \hat{\mathbf{\Sigma}}_\epsilon^{\mathrm{T}}, \tag{46}$$

where the right-hand side represents the transpose with respect to the computational basis. Then, we can rewrite the last line of (42) as

$$
\begin{aligned}
Q_\epsilon^* \left(\frac{\tilde{x}}{G}\right) P^{(\tau)} \left(\frac{\tilde{x}}{G}\right) &\propto \left\langle \tilde{x} \left| \sqrt{\frac{1}{Q_{\max}^{(\tau)}} \mathbf{Q}^{(\tau)}} \mathbf{F}_D^\dagger \hat{\mathbf{q}}^{(\rho)} \left(\frac{1}{Q_{\max}^{(\tau)}}\hat{\mathbf{\Sigma}} + \frac{\epsilon}{Q_{\max}^{(\tau)}}\mathbb{1}\right)^{-1} \mathbf{F}_D \sqrt{\frac{1}{Q_{\max}^{(\tau)}} \mathbf{Q}^{(\tau)}} \right| \tilde{x} \right\rangle \\
&= \left\langle \tilde{x} \left| \sqrt{\frac{1}{Q_{\max}^{(\tau)}} \mathbf{Q}^{(\tau)}} \mathbf{F}_D^\dagger \sqrt{\hat{\mathbf{q}}^{(\rho)}} \left(\hat{\mathbf{\Sigma}}_\epsilon^{(\rho)}\right)^{-1} \sqrt{\hat{\mathbf{q}}^{(\rho)}} \mathbf{F}_D \sqrt{\frac{1}{Q_{\max}^{(\tau)}} \mathbf{Q}^{(\tau)}} \right| \tilde{x} \right\rangle \\
&= \left\langle \tilde{x} \left| \sqrt{\frac{1}{Q_{\max}^{(\tau)}} \mathbf{Q}^{(\tau)}} \mathbf{F}_D^\dagger \sqrt{\hat{\mathbf{q}}^{(\rho)}} \hat{\mathbf{\Sigma}}_\epsilon^{-1} \sqrt{\hat{\mathbf{q}}^{(\rho)}} \mathbf{F}_D \sqrt{\frac{1}{Q_{\max}^{(\tau)}} \mathbf{Q}^{(\tau)}} \right| \tilde{x} \right\rangle,
\end{aligned} \tag{47}
$$

where this probability distribution is normalized by $\sum_{\tilde{x} \in \tilde{\mathcal{X}}} Q_\epsilon^* \left(\frac{\tilde{x}}{G}\right) P^{(\tau)}\left(\frac{\tilde{x}}{G}\right) = 1$.

To prove the proposition, we analyze the probability distribution obtained from the measurement of the quantum state

$$|\Psi\rangle^{XX'} \propto \sum_{\tilde{x} \in \tilde{\mathcal{X}}} \hat{\mathbf{\Sigma}}_\epsilon^{-\frac{1}{2}} |\tilde{x}\rangle^X \otimes \sqrt{\frac{1}{Q_{\max}^{(\tau)}} \mathbf{Q}^{(\tau)}} \mathbf{F}_D^\dagger \sqrt{\hat{\mathbf{q}}^{(\rho)}} |\tilde{x}\rangle^{X'} \in \mathcal{H}^X \otimes \mathcal{H}^{X'}. \tag{48}$$

where $\sqrt{\hat{\mathbf{q}}^{(\rho)}} |\tilde{x}\rangle^{X'} = \sqrt{\hat{q}^{(\rho)}(\tilde{x})} |\tilde{x}\rangle^{X'}$. For any operators $\mathbf{A}$ on $\mathcal{H}^X$ and $\mathbf{B}$ on $\mathcal{H}^{X'}$ where the dimensions of these Hilbert spaces are the same

$$\dim \mathcal{H}^X = \dim \mathcal{H}^{X'}, \tag{49}$$

a straightforward linear algebraic calculation shows [1]

$$\sum_{\tilde{x} \in \tilde{\mathcal{X}}} \mathbf{A} |\tilde{x}\rangle^X \otimes \mathbf{B} |\tilde{x}\rangle^{X'} = \sum_{\tilde{x} \in \tilde{\mathcal{X}}} |\tilde{x}\rangle^X \otimes \mathbf{B}\mathbf{A}^{\mathrm{T}} |\tilde{x}\rangle^{X'}. \tag{50}$$

Applying this equality to (48), we have

$$
\begin{aligned}
|\Psi\rangle^{XX'} &\propto \sum_{\tilde{x} \in \tilde{\mathcal{X}}} \hat{\mathbf{\Sigma}}_\epsilon^{-\frac{1}{2}} |\tilde{x}\rangle^X \otimes \sqrt{\frac{1}{Q_{\max}^{(\tau)}} \mathbf{Q}^{(\tau)}} \mathbf{F}_D^\dagger \sqrt{\hat{\mathbf{q}}^{(\rho)}} |\tilde{x}\rangle^{X'} \\
&= \sum_{\tilde{x} \in \tilde{\mathcal{X}}} |\tilde{x}\rangle^X \otimes \sqrt{\frac{1}{Q_{\max}^{(\tau)}} \mathbf{Q}^{(\tau)}} \mathbf{F}_D^\dagger \sqrt{\hat{\mathbf{q}}^{(\rho)}} \left(\hat{\mathbf{\Sigma}}_\epsilon^{\mathrm{T}}\right)^{-\frac{1}{2}} |\tilde{x}\rangle^{X'} \\
&= \sum_{\tilde{x} \in \tilde{\mathcal{X}}} |\tilde{x}\rangle^X \otimes \sqrt{\frac{1}{Q_{\max}^{(\tau)}} \mathbf{Q}^{(\tau)}} \mathbf{F}_D^\dagger \sqrt{\hat{\mathbf{q}}^{(\rho)}} \hat{\mathbf{\Sigma}}_\epsilon^{-\frac{1}{2}} |\tilde{x}\rangle^{X'},
\end{aligned} \tag{51}
$$

where the last line follows from (46). If we performed a measurement of $|\Psi\rangle^{XX'}$ in the computational basis $\left\{|\tilde{x}\rangle^X \otimes |\tilde{x}'\rangle^{X'}\right\}$, the probability distribution of measurement outcomes, i.e., the square of the amplitude as summarized in Sec. A, would be

$$
\begin{aligned}
p(\tilde{x}, \tilde{x}') &= \left| \left(\langle \tilde{x}|^X \otimes \langle \tilde{x}'|^{X'}\right) |\Psi\rangle^{XX'} \right|^2 \\
&\propto \left| \left\langle \tilde{x}' \left| \sqrt{\frac{1}{Q_{\max}^{(\tau)}} \mathbf{Q}^{(\tau)}} \mathbf{F}_D^\dagger \sqrt{\hat{\mathbf{q}}^{(\rho)}} \hat{\mathbf{\Sigma}}_\epsilon^{-\frac{1}{2}} \right| \tilde{x} \right\rangle \right|^2 \\
&= \left\langle \tilde{x}' \left| \sqrt{\frac{1}{Q_{\max}^{(\tau)}} \mathbf{Q}^{(\tau)}} \mathbf{F}_D^\dagger \sqrt{\hat{\mathbf{q}}^{(\rho)}} \hat{\mathbf{\Sigma}}_\epsilon^{-\frac{1}{2}} \right| \tilde{x} \right\rangle \left\langle \tilde{x} \left| \hat{\mathbf{\Sigma}}_\epsilon^{-\frac{1}{2}} \sqrt{\hat{\mathbf{q}}^{(\rho)}} \mathbf{F}_D \sqrt{\frac{1}{Q_{\max}^{(\tau)}} \mathbf{Q}^{(\tau)}} \right| \tilde{x}' \right\rangle.
\end{aligned} \tag{52}
$$

Since a measurement of the register $X'$ yields an outcome $\tilde{x}'$ with probability $p(\tilde{x}') = \sum_{\tilde{x} \in \tilde{\mathcal{X}}} p(\tilde{x}, \tilde{x}')$ as summarized in Sec. A, we obtain

$$
\begin{aligned}
p(\tilde{x}') &\propto \sum_{\tilde{x} \in \tilde{\mathcal{X}}} \left\langle \tilde{x}' \left| \sqrt{\frac{1}{Q_{\max}^{(\tau)}} \mathbf{Q}^{(\tau)}} \mathbf{F}_D^\dagger \sqrt{\hat{\mathbf{q}}^{(\rho)}} \hat{\mathbf{\Sigma}}_\epsilon^{-\frac{1}{2}} \right| \tilde{x} \right\rangle \left\langle \tilde{x} \left| \hat{\mathbf{\Sigma}}_\epsilon^{-\frac{1}{2}} \sqrt{\hat{\mathbf{q}}^{(\rho)}} \mathbf{F}_D \sqrt{\frac{1}{Q_{\max}^{(\tau)}} \mathbf{Q}^{(\tau)}} \right| \tilde{x}' \right\rangle \\
&= \left\langle \tilde{x}' \left| \sqrt{\frac{1}{Q_{\max}^{(\tau)}} \mathbf{Q}^{(\tau)}} \mathbf{F}_D^\dagger \sqrt{\hat{\mathbf{q}}^{(\rho)}} \hat{\mathbf{\Sigma}}_\epsilon^{-\frac{1}{2}} \left( \sum_{\tilde{x} \in \tilde{\mathcal{X}}} |\tilde{x}\rangle \langle \tilde{x}| \right) \hat{\mathbf{\Sigma}}_\epsilon^{-\frac{1}{2}} \sqrt{\hat{\mathbf{q}}^{(\rho)}} \mathbf{F}_D \sqrt{\frac{1}{Q_{\max}^{(\tau)}} \mathbf{Q}^{(\tau)}} \right| \tilde{x}' \right\rangle \\
&= \left\langle \tilde{x}' \left| \sqrt{\frac{1}{Q_{\max}^{(\tau)}} \mathbf{Q}^{(\tau)}} \mathbf{F}_D^\dagger \sqrt{\hat{\mathbf{q}}^{(\rho)}} \hat{\mathbf{\Sigma}}_\epsilon^{-\frac{1}{2}} \mathbb{1} \hat{\mathbf{\Sigma}}_\epsilon^{-\frac{1}{2}} \sqrt{\hat{\mathbf{q}}^{(\rho)}} \mathbf{F}_D \sqrt{\frac{1}{Q_{\max}^{(\tau)}} \mathbf{Q}^{(\tau)}} \right| \tilde{x}' \right\rangle \\
&= \left\langle \tilde{x}' \left| \sqrt{\frac{1}{Q_{\max}^{(\tau)}} \mathbf{Q}^{(\tau)}} \mathbf{F}_D^\dagger \sqrt{\hat{\mathbf{q}}^{(\rho)}} \hat{\mathbf{\Sigma}}_\epsilon^{-1} \sqrt{\hat{\mathbf{q}}^{(\rho)}} \mathbf{F}_D \sqrt{\frac{1}{Q_{\max}^{(\tau)}} \mathbf{Q}^{(\tau)}} \right| \tilde{x}' \right\rangle .
\end{aligned}
\tag{53}
$$

Recall that the normalization $\left\| |\Psi\rangle^{XX'} \right\|_2 = 1$ of the quantum state yields $\sum_{\tilde{x}' \in \tilde{\mathcal{X}}} p(\tilde{x}') = 1$. Therefore, (47) and (53) yield

$$
p(\tilde{x}) = Q_\epsilon^* \left( \frac{\tilde{x}}{G} \right) P^{(\tau)} \left( \frac{\tilde{x}}{G} \right),
\tag{54}
$$

which shows the conclusion. $\qquad\square$

# E    Quantum algorithm for sampling an optimized random feature

In this section, we show our quantum algorithm for sampling an optimized random feature and bound its runtime.

Algorithm 1 shows our quantum algorithm. Note that each line of Algorithm 1 is performed approximately with a sufficiently small precision to achieve the overall sampling precision $\Delta > 0$, in the same way as classical algorithms that deal with real number using fixed- or floating-point number representation with a sufficiently small precision. In Algorithm 1, we represent computation of the function $Q^{(\tau)}$ as a quantum oracle $\mathcal{O}_\tau$. This oracle $\mathcal{O}_\tau$ computes $Q^{(\tau)}$ while maintaining the superpositions (i.e., linear combinations) in a given quantum state, that is,

$$
\mathcal{O}_\tau \left( \sum_v \alpha_v |v\rangle \otimes |0\rangle \right) = \sum_v \alpha_v |v\rangle \otimes |Q^{(\tau)}(v)\rangle,
\tag{55}
$$

where $\alpha_v \in \mathbb{C}$ can be any coefficient of the given state, and $|v\rangle$ and $|Q^{(\tau)}(v)\rangle$ are computational-basis states corresponding to bit strings representing $v \in \mathcal{V}$ and $Q^{(\tau)}(v) \in \mathbb{R}$ in the fixed-point number representation with sufficient precision. We use $\mathcal{O}_\tau$ for simplicity of the presentation, and unlike $\mathcal{O}_\rho$ given by (5), $\mathcal{O}_\tau$ is not a black box in our quantum algorithm since we can implement $\mathcal{O}_\tau$ explicitly by a quantum circuit under the assumption in the main text, without using QRAM discussed in Sec. B. Using the assumption that classical computation can evaluate the function $Q^{(\tau)}$ efficiently in a short time denoted by

$$
T_\tau = O(\mathrm{poly}(D)),
\tag{56}
$$

we can efficiently implement $\mathcal{O}_\tau$ in runtime $O(T_\tau)$; in particular, if we can compute $Q^{(\tau)}$ by numerical libraries using arithmetics in runtime $T_\tau$, then a quantum computer can also perform the same arithmetics to implement $\mathcal{O}_\tau$ in runtime $O(T_\tau)$ [18]. Note that even if the numerical libraries evaluated $Q^{(\tau)}$ by means of a lookup table stored in RAM, quantum computers could instead use the QRAM. In the following, for simplicity of the presentation, the runtime of $\mathcal{O}_\tau$ per query may also be denoted by $T_\tau$ (with abuse of notation) as our runtime analysis ignores constant factors. Note that in the case of the Gaussian kernel and the Laplacian kernel, $Q^{(\tau)}$ is given in terms of special functions as shown in the main text, and we have

$$
T_\tau = O(D),
\tag{57}
$$

which satisfies (56).

In the rest of this section, we prove the following theorem that bounds the runtime of Algorithm 1.

**Theorem 1** (Runtime of our quantum algorithm for sampling an optimized random feature). *Given $D$-dimensional data discretized by $G > 0$, for any learning accuracy $\epsilon > 0$ and any sampling precision $\Delta > 0$, the runtime $T_1$ of Algorithm 1 for sampling each optimized random feature $v_G \in \mathcal{V}_G$ from a distribution $Q(v_G)P^{(\tau)}(v_G)$ close to the optimized distribution $Q_\epsilon^*(v_G)P^{(\tau)}(v_G)$ with precision $\sum_{v_G \in \mathcal{V}_G} |Q(v_G)P^{(\tau)}(v_G) - Q_\epsilon^*(v_G)P^{(\tau)}(v_G)| \leqq \Delta$ is*

$$T_1 = O\left(D\log(G)\log\log(G) + T_\rho + T_\tau\right) \times \widetilde{O}\left(\frac{Q_{\max}^{(\tau)}}{\epsilon}\,\mathrm{polylog}\left(\frac{1}{\Delta}\right)\right),$$

*where $T_\rho$ and $T_\tau$ are the runtime of the quantum oracles $\mathcal{O}_\rho$ and $\mathcal{O}_\tau$ per query, and $Q_{\max}^{(\tau)}$, $\mathcal{O}_\rho$ and $\mathcal{O}_\tau$ are defined as (29), (5), and (55), respectively.*

---

**Algorithm 1** Quantum algorithm for sampling an optimized random feature (quOptRF).

---

**Input:** A desired accuracy $\epsilon > 0$ in the supervised learning, sampling precision $\Delta > 0$, quantum oracles $\mathcal{O}_\rho$ in (5) and $\mathcal{O}_\tau$ in (55), and $Q_{\max}^{(\tau)} > 0$ in (29).

**Output:** An optimized random feature $v_G \in \mathcal{V}_G$ sampled from a probability distribution $Q(v_G)P^{(\tau)}(v_G)$ with $\sum_{v_G \in \mathcal{V}_G} |Q(v_G)P^{(\tau)}(v_G) - Q_\epsilon^*(v_G)P^{(\tau)}(v_G)| \leqq \Delta$.

1: Initialize quantum registers $X$ and $X'$, load data onto $X'$ by $\mathcal{O}_\rho$, and perform CNOT gates on $X$ and $X'$

$$|0\rangle^X \otimes |0\rangle^{X'} \xrightarrow{\mathcal{O}_\rho} \sum_{\tilde{x} \in \tilde{\mathcal{X}}} |0\rangle^X \otimes \sqrt{\hat{\mathbf{q}}^{(\rho)}}\,|\tilde{x}\rangle^{X'} \xrightarrow{\mathrm{CNOT}} \sum_{\tilde{x} \in \tilde{\mathcal{X}}} |\tilde{x}\rangle^X \otimes \sqrt{\hat{\mathbf{q}}^{(\rho)}}\,|\tilde{x}\rangle^{X'}. \tag{58}$$

2: Perform $\mathbf{F}_D^\dagger$ on $X'$ by QFT [19] to obtain

$$\sum_{\tilde{x} \in \tilde{\mathcal{X}}} |\tilde{x}\rangle^X \otimes \mathbf{F}_D^\dagger \sqrt{\hat{\mathbf{q}}^{(\rho)}}\,|\tilde{x}\rangle^{X'}. \tag{59}$$

3: Apply the block encoding of $\sqrt{\frac{1}{Q_{\max}^{(\tau)}}\mathbf{Q}^{(\tau)}}$ (Lemma 1) to $X'$ followed by amplitude amplification to obtain a state proportional to

$$\sum_{\tilde{x} \in \tilde{\mathcal{X}}} |\tilde{x}\rangle^X \otimes \sqrt{\frac{1}{Q_{\max}^{(\tau)}}\mathbf{Q}^{(\tau)}}\,\mathbf{F}_D^\dagger \sqrt{\hat{\mathbf{q}}^{(\rho)}}\,|\tilde{x}\rangle^{X'}. \tag{60}$$

4: Apply the block encoding of $\hat{\boldsymbol{\Sigma}}_\epsilon^{-\frac{1}{2}}$ (Lemma 2) to $X$ to obtain the quantum state $|\Psi\rangle^{XX'}$ in Proposition 2. {This step requires our technical contribution since *no assumption on sparsity and low rank is imposed on $\hat{\boldsymbol{\Sigma}}_\epsilon$.*}

5: Perform a measurement of $X'$ in the computational basis to obtain $\tilde{x}$ with probability $Q_\epsilon^*\left(\frac{\tilde{x}}{G}\right)P^{(\tau)}\left(\frac{\tilde{x}}{G}\right)$.

6: **Return** $v_G = \frac{\tilde{x}}{G}$.

---

To prove Theorem 1, in the following, we construct efficient implementations of block encodings; in particular, we first show a block encoding of $\sqrt{\frac{1}{Q_{\max}^{(\tau)}}\mathbf{Q}^{(\tau)}}$, and then using this block encoding, we show that of $\hat{\boldsymbol{\Sigma}}_\epsilon$. Then, we will provide the runtime analysis of Algorithm 1 using these block encodings. In Algorithm 1, we combine these block encodings with two fundamental subroutines of quantum algorithms, namely, quantum Fourier transform (QFT) [19, 20] and quantum singular value transformation (QSVT) [5]. Using QFT, we can implement the unitary operator $\mathbf{F}$ defined as (25) with precision $\Delta$ by a quantum circuit composed of $O\left(\log(G)\log\left(\frac{\log G}{\Delta}\right)\right)$ gates [19]. Thus, we can implement $\mathbf{F}_D = F^{\otimes D}$ defined as (26) by a quantum circuit composed of gates of order

$$O\left(D\log(G)\log\left(\frac{\log G}{\Delta}\right)\right). \tag{61}$$

Note that QFT in Ref. [19] that we use in the following analysis has slightly better runtime than QFT in Ref. [20] by a poly-logarithmic term in $\frac{1}{\Delta}$, but we may also use QFT in Ref. [20] without changing

any statement of our lemmas and theorems since this poly-logarithmic term is not dominant. In our analysis, we multiply two numbers represented by $O\left(\log\left(G\right)\right)$ bits using the algorithm shown in Ref. [21] within time

$$O\left(\log\left(G\right)\log\log\left(G\right)\right), \tag{62}$$

which we can perform also on quantum computer by implementing arithmetics using a quantum circuit [18]. Note that we could also use exact quantum Fourier transform [1] or grammar-school-method multiplication instead of these algorithms in Refs. [19, 21], to decrease a constant factor in the runtime of our algorithm at the expense of logarithmically increasing the asymptotic scaling in terms of $G$ from $\log\left(G\right)\log\log\left(G\right)$ to $\left(\log\left(G\right)\right)^2$. In the following runtime analysis, we use the definition (4) of block encoding to clarify the dependency on precision $\Delta$, and the quantum registers for storing real number use fixed-point number representation with sufficient precision $O(\Delta)$ to achieve the overall precision $\Delta$.

Our construction of block encodings of $\sqrt{\frac{1}{Q_{\max}^{(\tau)}}\mathbf{Q}^{(\tau)}}$ and $\hat{\boldsymbol{\Sigma}}_\epsilon$ is based on a prescription of constructing a block encoding from a quantum circuit for implementing a measurement described by a positive operator-valued measure (POVM) [5]. In particular, for any precision $\Delta > 0$ and any POVM operator $\boldsymbol{\Lambda}$, that is, an operator satisfying $0 \leqq \boldsymbol{\Lambda} \leqq \mathbb{1}$, let $\mathbf{U}$ be a unitary operator represented by a quantum circuit that satisfies for any state $|\psi\rangle$

$$\left|\mathrm{Tr}\left[|\psi\rangle\langle\psi|\,\boldsymbol{\Lambda}\right] - \mathrm{Tr}\left[\mathbf{U}\left(|0\rangle\langle0|^{\otimes n}\otimes|\psi\rangle\langle\psi|\right)\mathbf{U}^\dagger\left(|0\rangle\langle0|^{\otimes 1}\otimes\mathbb{1}\right)\right]\right| \leqq \Delta, \tag{63}$$

where $|0\rangle^{\otimes n}$ is a fixed state of $n$ auxiliary qubits. The quantum circuit $\mathbf{U}$ in (63) means that $\mathbf{U}$ implements a quantum measurement represented by the POVM operator $\boldsymbol{\Lambda}$ with precision $\Delta$; that is, given any input state $|\psi\rangle$ and $n$ auxiliary qubits initially prepared in $|0\rangle^{\otimes n}$, if we perform the circuit $\mathbf{U}$ to obtain a state $\mathbf{U}\left(|0\rangle^{\otimes n}\otimes|\psi\rangle\right)$ and perform a measurement of one of the qubits for the obtained state in the computational basis $\{|0\rangle, |1\rangle\}$, then we obtain a measurement outcome $0$ with probability

$$\mathrm{Tr}\left[\mathbf{U}\left(|0\rangle\langle0|^{\otimes n}\otimes|\psi\rangle\langle\psi|\right)\mathbf{U}^\dagger\left(|0\rangle\langle0|^{\otimes 1}\otimes\mathbb{1}\right)\right]. \tag{64}$$

Then, it is known that we can construct a $(1, 1+n, \Delta)$-block encoding of $\boldsymbol{\Lambda}$ using one $\mathbf{U}$, one $\mathbf{U}^\dagger$, and one quantum logic gate (i.e., the controlled NOT (CNOT) gate) [5]. The CNOT gate is defined as a two-qubit unitary operator

$$\mathrm{CNOT} \coloneqq |0\rangle\langle0|\otimes\mathbb{1} + |1\rangle\langle1|\otimes\sigma_x, \tag{65}$$

where the first qubit is a controlled qubit, the second qubit is a target qubit, and $\sigma_x$ is a Pauli unitary operator

$$\sigma_x \coloneqq |0\rangle\langle1| + |1\rangle\langle0|. \tag{66}$$

The CNOT gate acts as

$$\mathrm{CNOT}\left(\left(\alpha_0|0\rangle + \alpha_1|1\rangle\right)\otimes|0\rangle\right) = \alpha_0|0\rangle\otimes|0\rangle + \alpha_1|1\rangle\otimes|1\rangle. \tag{67}$$

For a given POVM operator $\boldsymbol{\Lambda}$, no general way of constructing the circuit representing $\mathbf{U}$ in (63) has been shown in Ref. [5]; in contrast, we here explicitly construct the circuit for a diagonal POVM operator $\boldsymbol{\Lambda} = \sqrt{\frac{1}{Q_{\max}^{(\tau)}}\mathbf{Q}^{(\tau)}}$ in the following lemma, using the quantum oracle $\mathcal{O}_\tau$. Note that since a diagonal operator is sparse, a conventional way of implementing the block encoding of a sparse operator [5] would also be applicable to construct a block encoding of $\sqrt{\frac{1}{Q_{\max}^{(\tau)}}\mathbf{Q}^{(\tau)}}$; however, our key contribution here is to use the circuit for the block encoding of $\sqrt{\frac{1}{Q_{\max}^{(\tau)}}\mathbf{Q}^{(\tau)}}$ as a building block of a more complicated block encoding, i.e., the block encoding of $\hat{\boldsymbol{\Sigma}}_\epsilon$, which is not necessarily sparse or of low rank.

**Lemma 1** (Block encoding of a diagonal POVM operator). *For any diagonal positive semidefinite operator $\mathbf{Q}^{(\tau)}$ defined as (24), we can implement a $\left(1, O\left(D\log\left(G\right)\mathrm{polylog}\left(\frac{1}{\Delta}\right)\right), \Delta\right)$-block encoding of $\sqrt{\frac{1}{Q_{\max}^{(\tau)}}\mathbf{Q}^{(\tau)}}$ by a quantum circuit composed of $O\left(D\log\left(G\right)\log\log\left(G\right)\mathrm{polylog}\left(\frac{1}{\Delta}\right)\right)$ gates and one query to the quantum oracle $\mathcal{O}_\tau$.*

Figure 1: A quantum circuit representing a unitary operator $\mathbf{U}$ that achieves (63) for $\mathbf{\Lambda} = \sqrt{\frac{1}{Q_{\max}^{(\tau)}} \mathbf{Q}^{(\tau)}}$, which can be used for implementing a block encoding of $\sqrt{\frac{1}{Q_{\max}^{(\tau)}} \mathbf{Q}^{(\tau)}}$. This circuit achieves the transformation of quantum states shown in a chain starting from (75). The last controlled gate represents $\mathbf{CR}$. Regarding the notations on quantum circuits, see, e.g., [1].

*Proof.* We construct a quantum circuit representing a unitary operator $\mathbf{U}$ that achieves (63) for $\mathbf{\Lambda} = \sqrt{\frac{1}{Q_{\max}^{(\tau)}} \mathbf{Q}^{(\tau)}}$. We write the input quantum state as

$$|\psi\rangle^X = \sum_{\tilde{v} \in \tilde{\mathcal{X}}} \alpha_{\tilde{v}} |\tilde{v}\rangle^X \in \mathcal{H}^X. \tag{68}$$

Define a function

$$\theta_1(q) := \arccos\left(q^{\frac{1}{4}}\right). \tag{69}$$

Define unitary operators $\mathbf{U}_G$, $\mathbf{U}_{Q_{\max}^{(\tau)}}$, and $\mathbf{U}_{\theta_1}$ acting as

$$\mathbf{U}_G : |x\rangle \otimes |0\rangle \xrightarrow{\mathbf{U}_G} |x\rangle \otimes \left|\frac{x}{G}\right\rangle, \tag{70}$$

$$\mathbf{U}_{Q_{\max}^{(\tau)}} : |x\rangle \otimes |0\rangle \xrightarrow{\mathbf{U}_{Q_{\max}^{(\tau)}}} |x\rangle \otimes \left|\frac{x}{Q_{\max}^{(\tau)}}\right\rangle, \tag{71}$$

$$\mathbf{U}_{\theta_1} : |x\rangle \otimes |0\rangle \xrightarrow{\mathbf{U}_{\theta_1}} |x\rangle \otimes |\theta_1(x)\rangle. \tag{72}$$

Let $\mathbf{R}_\theta$ denote a unitary operator representing a one-qubit rotation

$$\mathbf{R}_\theta := \begin{pmatrix} \cos\theta & -\sin\theta \\ \sin\theta & \cos\theta \end{pmatrix}, \tag{73}$$

and a controlled rotation $\mathbf{CR}$ is defined as

$$\mathbf{CR} = \sum_\theta |\theta\rangle\langle\theta| \otimes \mathbf{R}_\theta. \tag{74}$$

Using these notations, we show a quantum circuit representing $\mathbf{U}$ in Fig. 1. This circuit achieves the following transformation up to precision $\Delta$

$$|\psi\rangle \otimes |0\rangle \otimes |0\rangle \otimes |0\rangle \otimes |0\rangle \otimes |0\rangle \tag{75}$$

$$\xrightarrow{\mathbf{U}_G} \sum_{\tilde{v} \in \tilde{\mathcal{X}}} \alpha_{\tilde{v}} |\tilde{v}\rangle \otimes \left|\frac{\tilde{v}}{G}\right\rangle \otimes |0\rangle \otimes |0\rangle \otimes |0\rangle \otimes |0\rangle \tag{76}$$

$$\xrightarrow{\mathcal{O}_\tau} \sum_{\tilde{v} \in \tilde{\mathcal{X}}} \alpha_{\tilde{v}} |\tilde{v}\rangle \otimes \left|\frac{\tilde{v}}{G}\right\rangle \otimes \left|Q^{(\tau)}\left(\frac{\tilde{v}}{G}\right)\right\rangle \otimes |0\rangle \otimes |0\rangle \otimes |0\rangle \tag{77}$$

$$\xrightarrow{\mathbf{U}_{Q_{\max}^{(\tau)}}} \sum_{\tilde{v} \in \tilde{\mathcal{X}}} \alpha_{\tilde{v}} |\tilde{v}\rangle \otimes \left|\frac{\tilde{v}}{G}\right\rangle \otimes \left|Q^{(\tau)}\left(\frac{\tilde{v}}{G}\right)\right\rangle \otimes \left|\frac{Q^{(\tau)}\left(\frac{\tilde{v}}{G}\right)}{Q_{\max}^{(\tau)}}\right\rangle \otimes |0\rangle \otimes |0\rangle \tag{78}$$

$$\xrightarrow{\mathbf{U}_{\theta_1}} \sum_{\tilde{v} \in \tilde{\mathcal{X}}} \alpha_{\tilde{v}} |\tilde{v}\rangle \otimes \left|\frac{\tilde{v}}{G}\right\rangle \otimes \left|Q^{(\tau)}\left(\frac{\tilde{v}}{G}\right)\right\rangle \otimes \left|\frac{Q^{(\tau)}\left(\frac{\tilde{v}}{G}\right)}{Q_{\max}^{(\tau)}}\right\rangle \otimes \left|\theta_1\left(\frac{Q^{(\tau)}\left(\frac{\tilde{v}}{G}\right)}{Q_{\max}^{(\tau)}}\right)\right\rangle \otimes |0\rangle \tag{79}$$

$$\xrightarrow{\mathbf{CR}} \sum_{\tilde{v}\in\tilde{\mathcal{X}}} \alpha_{\tilde{v}} \left|\tilde{v}\right\rangle \otimes \left|\frac{\tilde{v}}{G}\right\rangle \otimes \left|Q^{(\tau)}\left(\frac{\tilde{v}}{G}\right)\right\rangle \otimes \left|\frac{Q^{(\tau)}\left(\frac{\tilde{v}}{G}\right)}{Q_{\max}^{(\tau)}}\right\rangle \otimes \left|\theta_1\left(\frac{Q^{(\tau)}\left(\frac{\tilde{v}}{G}\right)}{Q_{\max}^{(\tau)}}\right)\right\rangle \otimes$$

$$\left( \left(\frac{1}{Q_{\max}^{(\tau)}}Q^{(\tau)}\left(\frac{\tilde{v}}{G}\right)\right)^{\frac{1}{4}}\left|0\right\rangle + \sqrt{1 - \sqrt{\frac{1}{Q_{\max}^{(\tau)}}Q^{(\tau)}\left(\frac{\tilde{v}}{G}\right)}}\left|1\right\rangle \right), \tag{80}$$

where the quantum registers for storing real number use fixed-point number representation with sufficient precision $O(\Delta)$ to achieve the overall precision $\Delta$ in (63). In (76), each of the $D$ elements of the vector $\tilde{v}$ in the first quantum register is multiplied by $\frac{1}{G}$ using arithmetics, and the result is stored in the second quantum register. Since $\frac{1}{G}$ can be approximately represented with precision $O(\Delta)$ using $O\left(\log\left(\frac{1}{\Delta}\right)\right)$ bits, these $D$ multiplications take $O\left(D\log\left(G\right)\log\log\left(G\right)\text{polylog}\left(\frac{1}{\Delta}\right)\right)$ time due to (62), which is dominant. The runtime of the quantum oracle $\mathcal{O}_\tau$ queried in (77) is $T_\tau$. We can multiply $\frac{1}{Q_{\max}^{(\tau)}}$ in (78) and calculate the elementary function $\theta_1$ in (79) up to precision $O(\Delta)$ by arithmetics within time $O\left(\text{polylog}\left(\frac{1}{\Delta}\right)\right)$ [18]. In (80), we apply $\mathbf{CR}$ defined as (74) to the last qubit controlled by the second last quantum register, which uses $O\left(\text{polylog}\left(\frac{1}{\Delta}\right)\right)$ gates since $\left|\theta\right\rangle$ stored in the second last register consists of $O\left(\text{polylog}\left(\frac{1}{\Delta}\right)\right)$ qubits. The measurement of the last qubit of (80) in the computational basis $\{\left|0\right\rangle, \left|1\right\rangle\}$ yields the outcome $0$ with probability

$$\sum_{\tilde{v}\in\tilde{\mathcal{X}}} \left|\alpha_{\tilde{v}}\right|^2 \sqrt{\frac{1}{Q_{\max}^{(\tau)}}Q^{(\tau)}\left(\frac{\tilde{v}}{G}\right)} = \text{Tr}\left[\left|\psi\right\rangle\left\langle\psi\right|\sqrt{\frac{1}{Q_{\max}^{(\tau)}}\mathbf{Q}^{(\tau)}}\right], \tag{81}$$

which achieves (63) for $\mathbf{\Lambda} = \sqrt{\frac{1}{Q_{\max}^{(\tau)}}\mathbf{Q}^{(\tau)}}$ within the claimed runtime. $\qquad\square$

Using the block encoding of $\sqrt{\frac{1}{Q_{\max}^{(\tau)}}\mathbf{Q}^{(\tau)}}$ as a building block, we construct a block encoding of $\hat{\mathbf{\Sigma}}_\epsilon$ in the following. Note that while the following proposition provides a $\left(1, O\left(D\log\left(G\right)\text{polylog}\left(\frac{1}{\Delta}\right)\right), \Delta\right)$-block encoding of $\frac{1}{1+\left(\epsilon/Q_{\max}^{(\tau)}\right)}\hat{\mathbf{\Sigma}}_\epsilon$, this block encoding is equivalently a $\left(1 + \left(\epsilon/Q_{\max}^{(\tau)}\right), O\left(D\log\left(G\right)\text{polylog}\left(\frac{1}{\Delta}\right)\right), \left(1 + \left(\epsilon/Q_{\max}^{(\tau)}\right)\right)\Delta\right)$-block encoding of $\hat{\mathbf{\Sigma}}_\epsilon$ by definition. In implementing the block encoding of $\hat{\mathbf{\Sigma}}_\epsilon$, we use the quantum oracle $\mathcal{O}_\rho$ defined as (5) in addition to $\mathcal{O}_\tau$.

**Lemma 2** (Block encoding of $\hat{\mathbf{\Sigma}}_\epsilon$). *For any $\epsilon > 0$ and any operator $\hat{\mathbf{\Sigma}}_\epsilon$ given in the form of (45), we can implement a $\left(1, O\left(D\log\left(G\right)\text{polylog}\left(\frac{1}{\Delta}\right)\right), \Delta\right)$-block encoding of*

$$\frac{1}{1+\left(\epsilon/Q_{\max}^{(\tau)}\right)}\hat{\mathbf{\Sigma}}_\epsilon$$

*by a quantum circuit composed of $O\left(D\log\left(G\right)\log\log\left(G\right)\text{polylog}\left(\frac{1}{\Delta}\right)\right)$ gates, one query to the quantum oracle $\mathcal{O}_\rho^\dagger$, i.e., the inverse of $\mathcal{O}_\rho$, and one query to the quantum oracle $\mathcal{O}_\tau$.*

*Proof.* We construct a quantum circuit representing a unitary operator $\mathbf{U}$ that achieves (63) for $\mathbf{\Lambda} = \hat{\mathbf{\Sigma}}_\epsilon$. We use the same notations as those in the proof of Lemma 1 except the following notations. The input state is written in this proof as

$$\left|\psi\right\rangle^X = \sum_{\tilde{x}\in\tilde{\mathcal{X}}} \alpha_{\tilde{x}} \left|\tilde{x}\right\rangle^X \in \mathcal{H}^X. \tag{82}$$

Define functions

$$\theta_2(q) := \arccos\left(\sqrt{q}\right), \tag{83}$$

$$\theta_3(\epsilon) := \arccos\left(\sqrt{\frac{\epsilon/Q_{\max}^{(\tau)}}{1+\left(\epsilon/Q_{\max}^{(\tau)}\right)}}\right). \tag{84}$$

Figure 2: A quantum circuit representing a unitary operator $\mathbf{U}$ that achieves (63) for $\mathbf{\Lambda} = \hat{\mathbf{\Sigma}}_\epsilon$, which can be used for implementing a block encoding of $\hat{\mathbf{\Sigma}}_\epsilon$. The notations are the same as those in Fig. 1. The first gate acting on two quantum registers $X$ and $X'$ collectively represents CNOT gates acting transversally on each of the qubits of these registers. A part of this circuit sandwiched by two vertical dashed lines is the same as the corresponding part in Fig. 1. Additionally, the circuit performs a preprocessing of the input state before performing the part corresponding to Fig. 1, which achieves the transformation of quantum states shown in a chain starting from (87). Also, the circuit performs the final gates $\mathbf{U}_{\theta_2}$ and $\mathbf{U}_R$ after the part corresponding to Fig. 1, which are followed by a measurement described by the analysis starting from (90).

Define a unitary operator $\mathbf{U}_{\theta_2}$ acting as

$$\mathbf{U}_{\theta_2} : |x\rangle \otimes |0\rangle \xrightarrow{U_{\theta_2}} |x\rangle \otimes |\theta_2(x)\rangle. \tag{85}$$

Let $\mathbf{U}_\rho$ denote a unitary operator representing a quantum circuit for implementing the oracle $\mathcal{O}_\rho$. Then, the unitary operator representing its inverse $\mathcal{O}_\rho^\dagger$ is given by $\mathbf{U}_\rho^\dagger$. Define a unitary operator

$$\begin{aligned}
\mathbf{U}_R = &\left( \mathbb{1} \otimes \mathbb{1}^{X'} \otimes |0\rangle\langle 0|^A \otimes \mathbb{1}^B \right) \\
&+ \left( \mathbb{1} \otimes \left( \mathbb{1}^{X'} - |0\rangle\langle 0|^{X'} \right) \otimes |1\rangle\langle 1|^A \otimes \sigma_x^B \right) \\
&+ \left( \sum_\theta |\theta\rangle\langle\theta| \otimes |0\rangle\langle 0|^{X'} \otimes |1\rangle\langle 1|^A \otimes \mathbf{R}_\theta^B \right),
\end{aligned} \tag{86}$$

where the first quantum register may store a real number $\theta$ in the fixed-point number representation with sufficient precision to achieve the overall precision $\Delta$ in (63), the second quantum register $\mathcal{H}^{X'}$ is isomorphic to the quantum register $\mathcal{H}^X$, i.e., is composed of the same number of qubits as $\mathcal{H}^X$, the third quantum register $\mathcal{H}^A$ is one auxiliary qubit, and the fourth quantum register $\mathcal{H}^B$ is another auxiliary qubit. The operators $\sigma_x^B$ and $\mathbf{R}_\theta^B$ on $\mathcal{H}^B$ are defined as (66) and (73), respectively. If the state of $A$ is $|0\rangle^A$, the first term of (86) does not change the state on $B$, and if $|1\rangle^A$, the second and third terms of (86) act as follows: unless the state of $X'$ is $|0\rangle^{X'}$, $\sigma_x^B$ in the second term of (86) flips $|0\rangle^B$ to $|1\rangle^B$, and if the state of $X'$ is $|0\rangle^{X'}$, $\mathbf{R}_\theta^B$ in the third term of (86) acts in the same way as (80).

Using these notations, we show a quantum circuit representing $\mathbf{U}$ in Fig. 2. While a part of the circuit in Fig. 2 sandwiched by two vertical dashed lines is the same as the corresponding part in Fig. 1, this circuit additionally performs a preprocessing of the input state before performing the part corresponding to Fig. 1, and the final gates $\mathbf{U}_{\theta_2}$ and $\mathbf{U}_R$ in Fig. 2 after the part corresponding to Fig. 1 are also different. This preprocessing implements the following transformation with sufficient precision $O(\Delta)$ to achieve the overall precision $\Delta$

$$|\psi\rangle^X \otimes |0\rangle \otimes |0\rangle \otimes |0\rangle \otimes |0\rangle \otimes |0\rangle^{X'} \otimes |0\rangle^A \otimes |0\rangle^B \tag{87}$$

$$\xrightarrow{\text{CNOT}} \sum_{\tilde{x}\in\tilde{\mathcal{X}}} \alpha_{\tilde{x}} |\tilde{x}\rangle^X \otimes |0\rangle \otimes |0\rangle \otimes |0\rangle \otimes |0\rangle \otimes |\tilde{x}\rangle^{X'} \otimes |0\rangle^A \otimes |0\rangle^B \tag{88}$$

$$\xrightarrow{\mathbf{F}_D^\dagger \otimes \mathcal{O}_\rho^\dagger \otimes \mathbf{R}_{\theta_3(\epsilon)}} \sum_{\tilde{x}\in\tilde{\mathcal{X}}} \alpha_{\tilde{x}} \mathbf{F}_D^\dagger |\tilde{x}\rangle^X \otimes |0\rangle \otimes |0\rangle \otimes |0\rangle \otimes |0\rangle \otimes \mathbf{U}_\rho^\dagger |\tilde{x}\rangle^{X'} \otimes$$

$$\left( \sqrt{\frac{\epsilon/Q_{\max}^{(\tau)}}{1 + \left(\epsilon/Q_{\max}^{(\tau)}\right)}} \, |0\rangle^A + \sqrt{\frac{1}{1 + \left(\epsilon/Q_{\max}^{(\tau)}\right)}} \, |1\rangle^A \right) \otimes |0\rangle^B . \qquad (89)$$

In (88), we use $O\left(D\log\left(G\right)\right)$ CNOT gates acting on each of the $O\left(D\log\left(G\right)\right)$ qubits of the quantum registers $X$ and $X'$. In (89), $\mathbf{F}_D^\dagger$ is implemented by $O\left(D\log\left(G\right)\log\left(\frac{\log G}{\Delta}\right)\right)$ gates as shown in (61), $\mathcal{O}_\rho^\dagger$ takes time $T_\rho$, and a fixed one-qubit rotation $\mathbf{R}_{\theta_3(\epsilon)}$ defined as (73) is implemented with precision $O(\Delta)$ using $O\left(\mathrm{polylog}\left(\frac{1}{\Delta}\right)\right)$ gates [1].

Then, after performing the same part as in Fig. 1, which is dominant, the circuit in Fig. 2 performs $\mathbf{U}_{\theta_2}$ defined as (85) and $\mathbf{U}_R$ defined as (86). We can implement $\mathbf{U}_{\theta_2}$ in the same way as (79), i.e., $\mathbf{U}_{\theta_1}$ in Lemma 1, using $O\left(\mathrm{polylog}\left(\frac{1}{\Delta}\right)\right)$ gates. We can implement $\mathbf{U}_R$ using $O\left(D\log\left(G\right)\mathrm{polylog}\left(\frac{1}{\Delta}\right)\right)$ gates since $|\theta\rangle$ is stored in $O\left(\mathrm{polylog}\left(\frac{1}{\Delta}\right)\right)$ qubits and $\mathcal{H}^{X'}$ consists of $O\left(D\log\left(G\right)\right)$ qubits.

After performing $\mathbf{U}_R$, we perform a measurement of the last qubit $\mathcal{H}^B$ in the computational basis $\{|0\rangle^B, |1\rangle^B\}$. To calculate the probability of obtaining the outcome $0$ in this measurement of $\mathcal{H}^B$, suppose that we performed a measurement of the one-qubit register $\mathcal{H}^A$ in the computational basis $\{|0\rangle^A, |1\rangle^A\}$. Then, we would obtain the outcome $|0\rangle^A$ with probability $\frac{\epsilon/Q_{\max}^{(\tau)}}{1 + \left(\epsilon/Q_{\max}^{(\tau)}\right)}$, and the outcome $|1\rangle^A$ with probability $\frac{1}{1 + \left(\epsilon/Q_{\max}^{(\tau)}\right)}$. Conditioned on the outcome $|0\rangle^A$, the measurement of $\mathcal{H}^B$ yields the outcome $|0\rangle^B$ with probability $1$ correspondingly to the first term of (86). Conditioned on $|1\rangle^A$, owing to the third term of (86), the measurement of $\mathcal{H}^B$ yields the outcome $|0\rangle^B$ with probability

$$\sum_{\tilde{x}' \in \tilde{\mathcal{X}}} \left| \sum_{\tilde{x} \in \tilde{\mathcal{X}}} \alpha_{\tilde{x}} \left\langle \tilde{x}' \left| \mathbf{F}_D^\dagger \right| \tilde{x} \right\rangle \langle 0 | \mathbf{U}_\rho^\dagger | \tilde{x} \rangle \sqrt{\frac{1}{Q_{\max}^{(\tau)}} Q^{(\tau)}\left(\frac{\tilde{x}'}{G}\right)} \right|^2$$

$$= \sum_{\tilde{x}' \in \tilde{\mathcal{X}}} \frac{1}{Q_{\max}^{(\tau)}} Q^{(\tau)}\left(\frac{\tilde{x}'}{G}\right) \left| \sum_{\tilde{x} \in \tilde{\mathcal{X}}} \alpha_{\tilde{x}} \left\langle \tilde{x}' \left| \mathbf{F}_D^\dagger \right| \tilde{x} \right\rangle \langle 0 | \mathbf{U}_\rho^\dagger | \tilde{x} \rangle \right|^2 . \qquad (90)$$

Note that the second term of (86) has no contribution in (90) because $\sigma_x^B$ flips $|0\rangle^B$ to $|1\rangle^B$. Due to $\left| \langle 0 | \mathbf{U}_\rho^\dagger | \tilde{x} \rangle \right| = \left| \langle \tilde{x} | \left( \sum_{\tilde{x}'} \sqrt{\hat{q}^{(\rho)}(\tilde{x}')} |\tilde{x}'\rangle \right) \right| = \sqrt{\hat{q}^{(\rho)}(\tilde{x})}$, we have

$$(90) = \sum_{\tilde{x}' \in \tilde{\mathcal{X}}} \frac{1}{Q_{\max}^{(\tau)}} Q^{(\tau)}\left(\frac{\tilde{x}'}{G}\right) \left| \sum_{\tilde{x} \in \tilde{\mathcal{X}}} \alpha_{\tilde{x}} \left\langle \tilde{x}' \left| \mathbf{F}_D^\dagger \right| \tilde{x} \right\rangle \sqrt{\hat{q}^{(\rho)}(\tilde{x})} \right|^2$$

$$= \sum_{\tilde{x}' \in \tilde{\mathcal{X}}} \frac{1}{Q_{\max}^{(\tau)}} Q^{(\tau)}\left(\frac{\tilde{x}'}{G}\right) \left| \sum_{\tilde{x} \in \tilde{\mathcal{X}}} \alpha_{\tilde{x}} \left\langle \tilde{x}' \left| \mathbf{F}_D^\dagger \sqrt{\hat{\mathbf{q}}^{(\rho)}} \right| \tilde{x} \right\rangle \right|^2 . \qquad (91)$$

By definition (82) of $|\psi\rangle$, we have

$$(91) = \sum_{\tilde{x}' \in \tilde{\mathcal{X}}} \frac{1}{Q_{\max}^{(\tau)}} Q^{(\tau)}\left(\frac{\tilde{x}'}{G}\right) \left| \left\langle \tilde{x}' \left| \mathbf{F}_D^\dagger \sqrt{\hat{\mathbf{q}}^{(\rho)}} \right| \psi \right\rangle \right|^2$$

$$= \sum_{\tilde{x}' \in \tilde{\mathcal{X}}} \frac{1}{Q_{\max}^{(\tau)}} Q^{(\tau)}\left(\frac{\tilde{x}'}{G}\right) \left\langle \tilde{x}' \left| \mathbf{F}_D^\dagger \sqrt{\hat{\mathbf{q}}^{(\rho)}} \right| \psi \right\rangle \langle \psi | \sqrt{\hat{\mathbf{q}}^{(\rho)}} \mathbf{F}_D \left| \tilde{x}' \right\rangle$$

$$= \frac{1}{Q_{\max}^{(\tau)}} \mathrm{Tr}\left[ \mathbf{F}_D^\dagger \sqrt{\hat{\mathbf{q}}^{(\rho)}} |\psi\rangle \langle \psi| \sqrt{\hat{\mathbf{q}}^{(\rho)}} \mathbf{F}_D \left( \sum_{\tilde{x}' \in \tilde{\mathcal{X}}} Q^{(\tau)}\left(\frac{\tilde{x}'}{G}\right) |\tilde{x}'\rangle \langle \tilde{x}'| \right) \right] . \qquad (92)$$

By definition (24) of $\mathbf{Q}^{(\tau)}$, we obtain

$$(92) = \frac{1}{Q_{\max}^{(\tau)}} \mathrm{Tr}\left[ \mathbf{F}_D^\dagger \sqrt{\hat{\mathbf{q}}^{(\rho)}} |\psi\rangle \langle \psi| \sqrt{\hat{\mathbf{q}}^{(\rho)}} \mathbf{F}_D \mathbf{Q}^{(\tau)} \right]$$

$$= \frac{1}{Q_{\max}^{(\tau)}} \operatorname{Tr}\left[|\psi\rangle\langle\psi| \sqrt{\hat{\mathbf{q}}^{(\rho)}} \mathbf{F}_D \mathbf{Q}^{(\tau)} \mathbf{F}_D^\dagger \sqrt{\hat{\mathbf{q}}^{(\rho)}}\right]$$

$$= \frac{1}{Q_{\max}^{(\tau)}} \operatorname{Tr}\left[|\psi\rangle\langle\psi| \sqrt{\hat{\mathbf{q}}^{(\rho)}} \mathbf{k} \sqrt{\hat{\mathbf{q}}^{(\rho)}}\right], \tag{93}$$

where the last equality follows from the perfect reconstruction of the kernel $k$ shown in Proposition 1. Therefore, since a measurement of the auxiliary qubit $\mathcal{H}^A$ in the computational basis $\left\{|0\rangle^A, |1\rangle^A\right\}$ yields outcome 0 and 1 with probability $\frac{\epsilon/Q_{\max}^{(\tau)}}{1+\left(\epsilon/Q_{\max}^{(\tau)}\right)}$ and $\frac{1}{1+\left(\epsilon/Q_{\max}^{(\tau)}\right)}$ respectively, the circuit in Fig. 2 yields the outcome 0 with probability

$$\frac{\epsilon/Q_{\max}^{(\tau)}}{1+\left(\epsilon/Q_{\max}^{(\tau)}\right)} \times 1 + \frac{1}{1+\left(\epsilon/Q_{\max}^{(\tau)}\right)} \times \left(\frac{1}{Q_{\max}^{(\tau)}} \operatorname{Tr}\left[|\psi\rangle\langle\psi| \sqrt{\hat{\mathbf{q}}^{(\rho)}} \mathbf{k} \sqrt{\hat{\mathbf{q}}^{(\rho)}}\right]\right)$$

$$= \frac{1}{1+\left(\epsilon/Q_{\max}^{(\tau)}\right)} \times \operatorname{Tr}\left[|\psi\rangle\langle\psi| \left(\frac{\epsilon}{Q_{\max}^{(\tau)}}\mathbb{1}\right)\right]$$

$$\quad + \frac{1}{1+\left(\epsilon/Q_{\max}^{(\tau)}\right)} \times \operatorname{Tr}\left[|\psi\rangle\langle\psi| \left(\frac{1}{Q_{\max}^{(\tau)}} \sqrt{\hat{\mathbf{q}}^{(\rho)}} \mathbf{k} \sqrt{\hat{\mathbf{q}}^{(\rho)}}\right)\right]$$

$$= \frac{1}{1+\left(\epsilon/Q_{\max}^{(\tau)}\right)} \times \operatorname{Tr}\left[|\psi\rangle\langle\psi| \left(\frac{1}{Q_{\max}^{(\tau)}} \sqrt{\hat{\mathbf{q}}^{(\rho)}} \mathbf{k} \sqrt{\hat{\mathbf{q}}^{(\rho)}} + \frac{\epsilon}{Q_{\max}^{(\tau)}}\mathbb{1}\right)\right]$$

$$= \operatorname{Tr}\left[|\psi\rangle\langle\psi| \left(\frac{1}{1+\left(\epsilon/Q_{\max}^{(\tau)}\right)}\hat{\boldsymbol{\Sigma}}_\epsilon\right)\right], \tag{94}$$

where the last equality follows from the definition (45) of $\hat{\boldsymbol{\Sigma}}_\epsilon$, which achieves (63) for $\boldsymbol{\Lambda} = \frac{1}{1+\left(\epsilon/Q_{\max}^{(\tau)}\right)}\hat{\boldsymbol{\Sigma}}_\epsilon$ within a claimed runtime. $\qquad\square$

Using the block encodings in Lemmas 1 and 2, we prove Theorem 1 as follows.

*Proof of Theorem 1.* We prove that Algorithm 1 has the claimed runtime guarantee. The dominant step of Algorithm 1 is Step 5, as shown in the following.

In Step 2, after the initialization of $|0\rangle^X \otimes |0\rangle^{X'}$, we prepare $\sum_{\tilde{x}\in\tilde{\mathcal{X}}} |0\rangle^X \otimes \sqrt{\hat{\mathbf{q}}^{(\rho)}} |\tilde{x}\rangle^{X'}$ by one query to the oracle $\mathcal{O}_\rho$ defined as (5), followed by $O\left(D\log\left(G\right)\right)$ CNOT gates to prepare $\sum_{\tilde{x}\in\tilde{\mathcal{X}}} |\tilde{x}\rangle^X \otimes \sqrt{\hat{\mathbf{q}}^{(\rho)}} |\tilde{x}\rangle^{X'}$, since $\mathcal{H}^X$ consists of $O\left(D\log\left(G\right)\right)$ qubits. Step 3 performs $\mathbf{F}_D^\dagger$, which is implemented using $O\left(D\log\left(G\right)\log\left(\frac{\log G}{\Delta}\right)\right)$ gates as shown in (61). Step 4 is implemented by the block encoding of $\sqrt{\frac{1}{Q_{\max}^{(\tau)}}\mathbf{Q}^{(\tau)}}$ within time $O\left(D\log\left(G\right)\log\log\left(G\right)\operatorname{polylog}\left(\frac{1}{\Delta}\right)+T_\tau\right)$ as shown in Lemma 1. The runtime at this moment is $O\left(D\log\left(G\right)\log\log\left(G\right)\operatorname{polylog}\left(\frac{1}{\Delta}\right)+T_\rho+T_\tau\right)$. After applying the block encoding of $\sqrt{\frac{1}{Q_{\max}^{(\tau)}}\mathbf{Q}^{(\tau)}}$, we obtain a quantum state represented as a linear combination including a term

$$\sum_{\tilde{x}\in\tilde{\mathcal{X}}} |\tilde{x}\rangle^X \otimes \sqrt{\frac{1}{Q_{\max}^{(\tau)}}\mathbf{Q}^{(\tau)}} \mathbf{F}_D^\dagger \sqrt{\hat{\mathbf{q}}^{(\rho)}} |\tilde{x}\rangle^{X'}, \tag{95}$$

and the norm of this term is

$$\left\|\sum_{\tilde{x}\in\tilde{\mathcal{X}}} |\tilde{x}\rangle^X \otimes \sqrt{\frac{1}{Q_{\max}^{(\tau)}}\mathbf{Q}^{(\tau)}} \mathbf{F}_D^\dagger \sqrt{\hat{\mathbf{q}}^{(\rho)}} |\tilde{x}\rangle^{X'}\right\|_2$$

$$= \sqrt{\frac{\text{Tr}\left[\sqrt{\hat{\mathbf{q}}^{(\rho)}}\mathbf{F}_D\mathbf{Q}^{(\tau)}\mathbf{F}_D^\dagger\sqrt{\hat{\mathbf{q}}^{(\rho)}}\right]}{Q_{\max}^{(\tau)}}}$$

$$= \sqrt{\frac{\text{Tr}\left[\mathbf{F}_D\mathbf{Q}^{(\tau)}\mathbf{F}_D^\dagger\hat{\mathbf{q}}^{(\rho)}\right]}{Q_{\max}^{(\tau)}}}$$

$$= \sqrt{\frac{\text{Tr}\,\hat{\boldsymbol{\Sigma}}}{Q_{\max}^{(\tau)}}}, \tag{96}$$

where the last equality uses $\hat{\boldsymbol{\Sigma}} = \mathbf{k}\hat{\mathbf{q}}^{(\rho)} = \mathbf{F}_D\mathbf{Q}^{(\tau)}\mathbf{F}_D^\dagger\hat{\mathbf{q}}^{(\rho)}$ obtained from Proposition 1. For any translation-invariant kernel $\tilde{k}(x', x) = \tilde{k}_{\text{TI}}(x' - x)$, we can evaluate $\text{Tr}\,\hat{\boldsymbol{\Sigma}}$ as

$$\text{Tr}\,\hat{\boldsymbol{\Sigma}} = \text{Tr}\left[\mathbf{k}\hat{\mathbf{q}}^{(\rho)}\right] = \tilde{k}_{\text{TI}}(0)\,\text{Tr}\,\hat{\mathbf{q}}^{(\rho)} = \tilde{k}(0,0) = \Omega(1), \tag{97}$$

where we use the assumption $\tilde{k}(0,0) = \Omega(k(0,0)) = \Omega(1)$. Thus, to obtain the normalized quantum state proportional to the term (95), Step 4 is followed by amplitude amplification [6] that repeats the above steps $O\left(\sqrt{\frac{Q_{\max}^{(\tau)}}{\text{Tr}\,\hat{\boldsymbol{\Sigma}}}}\right) = O\left(\sqrt{Q_{\max}^{(\tau)}}\right)$ times. Therefore, at the end of Step 4 including the amplitude amplification, the runtime is

$$O\left(\left(\left(D\log(G)\log\log(G)\,\text{polylog}\left(\frac{1}{\Delta}\right) + T_\rho + T_\tau\right) \times \sqrt{Q_{\max}^{(\tau)}}\right)\right). \tag{98}$$

Step 5 is performed by implementing a block encoding of $\hat{\boldsymbol{\Sigma}}_\epsilon^{-\frac{1}{2}}$, which is obtained from quantum singular value transformation (QSVT) [5] of the block encoding of $\frac{1}{1+(\epsilon/Q_{\max}^{(\tau)})}\hat{\boldsymbol{\Sigma}}_\epsilon$ constructed in Lemma 2. The block encoding of $\frac{1}{1+(\epsilon/Q_{\max}^{(\tau)})}\hat{\boldsymbol{\Sigma}}_\epsilon$ can be implemented in time $O\left(D\log(G)\log\log(G)\,\text{polylog}\left(\frac{1}{\Delta}\right) + T_\rho + T_\tau\right)$ as shown in Lemma 2. Then, the QSVT combined with variable-time amplitude amplification [4, 22, 23] yields a block encoding of $\left(\frac{1}{1+(\epsilon/Q_{\max}^{(\tau)})}\hat{\boldsymbol{\Sigma}}_\epsilon\right)^{-\frac{1}{2}}$, which can be applied to any given quantum state up to $\Delta$ precision using the block encoding of $\frac{1}{1+(\epsilon/Q_{\max}^{(\tau)})}\hat{\boldsymbol{\Sigma}}_\epsilon$ repeatedly $\widetilde{O}\left(\left(\frac{Q_{\max}^{(\tau)}}{\epsilon} + 1\right)\text{polylog}\left(\frac{1}{\Delta}\right)\right)$ times [5]. This repetition includes the runtime required for the amplitude amplification, and $\frac{Q_{\max}^{(\tau)}}{\epsilon} + 1$ is the condition number of $\frac{1}{1+(\epsilon/Q_{\max}^{(\tau)})}\hat{\boldsymbol{\Sigma}}_\epsilon$ since it holds that

$$\frac{1}{1+(\epsilon/Q_{\max}^{(\tau)})}\frac{\epsilon}{Q_{\max}^{(\tau)}}\mathbb{1} \leqq \frac{1}{1+(\epsilon/Q_{\max}^{(\tau)})}\hat{\boldsymbol{\Sigma}}_\epsilon \leqq \mathbb{1}. \tag{99}$$

Thus, Step 5 including amplitude amplification can be implemented in time

$$O\left(D\log(G)\log\log(G)\,\text{polylog}\left(\frac{1}{\Delta}\right) + T_\rho + T_\tau\right) \times \widetilde{O}\left(\left(\frac{Q_{\max}^{(\tau)}}{\epsilon} + 1\right)\text{polylog}\left(\frac{1}{\Delta}\right)\right)$$

$$= O\left(D\log(G)\log\log(G) + T_\rho + T_\tau\right) \times \widetilde{O}\left(\frac{Q_{\max}^{(\tau)}}{\epsilon}\,\text{polylog}\left(\frac{1}{\Delta}\right)\right). \tag{100}$$

Therefore from (98) and (100), we obtain the total runtime at the end of Step 5 including amplitude amplification

$$O\left(D\log(G)\log\log(G) + T_\rho + T_\tau\right) \times \widetilde{O}\left(\left(\sqrt{Q_{\max}^{(\tau)}} + \frac{Q_{\max}^{(\tau)}}{\epsilon}\right)\text{polylog}\left(\frac{1}{\Delta}\right)\right)$$

$$= O\left(D\log(G)\log\log(G) + T_\rho + T_\tau\right) \times \widetilde{O}\left(\frac{Q_{\max}^{(\tau)}}{\epsilon}\,\text{polylog}\left(\frac{1}{\Delta}\right)\right), \tag{101}$$

which yields the conclusion. $\square$

# F Overall runtime of learning with optimized random features

In this section, we show the algorithm for the learning with optimized random features and bound its runtime.

Algorithm 2 shows the whole algorithm that achieves the learning using the optimized random features. In Algorithm 2, we sample the optimized random features efficiently by Algorithm 1, and then perform stochastic gradient descent (SGD) [24] shown in Algorithm 3. As explained in the main text, the SGD achieves linear regression to obtain coefficients of the estimate $\hat{f}_{M,v_m,\alpha_m} = \sum_{m=0}^{M-1} \alpha_m \varphi(v_m, \cdot) \approx f$, i.e.,

$$\alpha = \begin{pmatrix} \alpha_0 \\ \vdots \\ \alpha_{M-1} \end{pmatrix} \in \mathbb{R}^M, \tag{102}$$

where the optimal coefficient $\alpha$ minimizes the generalization error

$$I(\alpha) := \sum_{\tilde{x} \in \tilde{\mathcal{X}}} p^{(\rho)}(\tilde{x}) \left| f(\tilde{x}) - \sum_{m=0}^{M-1} \alpha_m \varphi(v_m, \tilde{x}) \right|^2, \tag{103}$$

and the examples of data are IID sampled according to $p^{(\rho)}(\tilde{x}) := \int_{\Delta_{\tilde{x}}} d\rho(x)$.

We remark that rather than the linear regression based on least-squares of $I$ in (103), Bach [25] analyzes regularized least-squares regression exploiting $Q_\epsilon^*$, but it may be hard to compute description of $Q_\epsilon^*$. To circumvent this hardness of using $Q_\epsilon^*$ in regularization, we could replace the regularization in Ref. [25] with $L_2$ regularization $R(\alpha) = \lambda \|\alpha\|_2^2$. Then, due to strong convexity [24], SGD reducing $I + R$ to $O(\epsilon)$ terminates after $O(\frac{1}{\epsilon\lambda})$ iterations, while further research is needed to clarify how the $L_2$ regularization affects the learning accuracy compared to minimizing $I$ without this regularization. In this paper, we consider the linear regression minimizing $I$ to simplify the analysis of the required runtime for achieving the desired learning accuracy.

We prove the following theorem that bounds the runtime of Algorithm 2. The sketch of the proof is as follows. To bound the runtime of Algorithm 2, we show that the required number $T$ of iterations for the SGD [24], i.e., Algorithm 3, to return $\alpha$ minimizing $I$ to accuracy $O(\epsilon)$ with high probability greater than $1 - \delta$ is

$$T = O\left(\frac{1}{\epsilon^2 Q_{\min}^2} \log\left(\frac{1}{\delta}\right)\right), \tag{104}$$

where $Q_{\min}$ is the minimum of $Q(v_0), \ldots, Q(v_{M-1})$ in Theorem 1

$$Q_{\min} := \min\{Q(v_m) : m \in \{0, \ldots, M-1\}\}, \tag{105}$$

and the parameter region $\mathcal{W}$ of $\alpha$ in Algorithm 3 is chosen as an $M$-dimensional ball of center $0$ and of radius $O\left(\frac{1}{\sqrt{MQ_{\min}}}\right)$. Note that step sizes used for SGD [24] shown in Algorithm 3 are chosen depending on the number $T$ of iterations so as to achieve (104), but we can also use step sizes independent of $T$ at expense of as small as poly-logarithmic slowdown in terms of $\epsilon$, $Q_{\min}$ compared to (104) [26]. In the $t$th iteration of the SGD for each $t \in \{1, \ldots, T\}$, we calculate an unbiased estimate $\hat{g}^{(t)}$ of the gradient $\nabla I$. Using the $t$th IID sampled data $(\tilde{x}_t, y_t)$ and a uniformly sampled random integer $m \in \{0, \ldots, M-1\}$, we show that we can calculate $\hat{g}^{(t)}$ within time $O(MD)$ in addition to one query to each of the classical oracles

$$\mathcal{O}_{\tilde{x}}(n) = \tilde{x}_n, \quad \mathcal{O}_y(n) = y_n \tag{106}$$

to get $(\tilde{x}_t, y_t)$ in time $T_{\tilde{x}}$ and $T_y$, respectively; that is, the runtime per iteration of the SGD is $O(MD + T_{\tilde{x}} + T_y)$. Combining Algorithm 1 with this SGD, we achieve the learning by Algorithm 2 within the following overall runtime.

**Theorem 2** (Overall runtime of learning with optimized random features)**.** *The runtime $T_2$ of Algorithm 2 for learning with optimized random features is*

$$T_2 = O(MT_1) + O\left((MD + T_{\tilde{x}} + T_y) \frac{1}{\epsilon^2 Q_{\min}^2} \log\left(\frac{1}{\delta}\right)\right),$$

*where $T_1$ appears in Theorem 1, the first term is the runtime of sampling $M$ optimized random features by Algorithm 1, and the second term is runtime of the SGD.*

*Remark* 2 (Omission of some parameters from the informal statement of Theorem 2 in the main text). In the informal statement of Theorem 2 in the main text, we omit the dependency on $Q_{\min}$ and $\frac{1}{\delta}$ from the runtime. In the parameter region of sampling optimized random features that are weighted by importance and that nearly minimize the required number $M$ of features, the minimal weight $Q_{\min}$ of these features is expected to be sufficiently large compared to $\epsilon$, not dominating the runtime, while we include $Q_{\min}$ in our runtime analysis in Supplementary Material to bound the worst-case runtime. In addition, the dependency on $\frac{1}{\delta}$ is logarithmic in Theorem 2. For these reasons, we simplify the presentation in the main text by omitting $Q_{\min}$ and $\frac{1}{\delta}$.

---

**Algorithm 2** Algorithm for learning with optimized random features.

---

**Input:** Inputs to Algorithms 1 and 3, required number $M$ of features for achieving the learning to accuracy $O(\epsilon)$.
**Output:** Optimized random features $v_0, \ldots, v_{M-1}$ and coefficients $\alpha_0, \ldots, \alpha_{M-1}$ for $\sum_m \alpha_m \varphi(v_m, \cdot)$ to achieve the learning to accuracy $O(\epsilon)$ with probability greater than $1 - \delta$.
 1: **for** $m \in \{0, \ldots, M-1\}$ **do**
 2:    $v_m \leftarrow$ quOptRF. {by Algorithm 1.}
 3: **end for**
 4: Minimize $I(\alpha)$ to accuracy $O(\epsilon)$ by SGD to obtain $\alpha_0, \ldots, \alpha_{M-1}$. {by Algorithm 3.}
 5: **Return** $v_0, \ldots, v_{M-1}, \alpha_0, \ldots, \alpha_{M-1}$.

---

**Algorithm 3** Stochastic gradient descent (SGD).

---

**Input:** A function $I : \mathcal{W} \to \mathbb{R}$, a projection $\Pi$ to a convex parameter region $\mathcal{W} \subset \mathbb{R}^M$ specified by $Q_{\min}$ in (105), number of iterations $T \in \mathbb{N}$ specified by (104), an initial point $\alpha^{(1)} \in \mathcal{W}$, $T$-dependent hyperparameters representing step sizes $\left(\eta^{(t)} : t = 1, \ldots, T\right)$ given in Ref. [24], classical oracle functions $\mathcal{O}_{\tilde{x}}, \mathcal{O}_y$ in (106) for calculating $\hat{g}^{(t)}$.
**Output:** Approximate solution $\alpha$ minimizing $I(\alpha)$.
 1: **for** $t \in \{1, \ldots, T\}$ **do**
 2:    Calculate an unbiased estimate $\hat{g}^{(t)}$ of the gradient of $I$ satisfying $\mathbb{E}\left[\hat{g}^{(t)}\right] = \nabla I(\alpha^{(t)})$.
 3:    $\alpha^{(t+1)} \leftarrow \Pi(\alpha^{(t)} - \eta^{(t)} \hat{g}^{(t)})$.
 4: **end for**
 5: **Return** $\alpha \leftarrow \alpha^{(T+1)}$.

---

*Proof.* We bound the runtime of each step of Algorithm 2. In Step 2, using Algorithm 1 repeatedly $M$ times, we obtain $M$ optimized random features within time

$$O(MT_1), \tag{107}$$

where $T_1$ is the runtime of Algorithm 1 given by Theorem 1. As for Step 4, we bound the runtime of the SGD in Algorithm 3. In the following, we show that the runtime of each iteration of the SGD is $O\left(MD + T_{\tilde{x}} + T_y\right)$, and the required number of iterations in the SGD is upper bounded by $O\left(\frac{1}{\epsilon^2 Q_{\min}^2} \log\left(\frac{1}{\delta}\right)\right)$.

We analyze the runtime of each iteration of the SGD. The dominant step in the $t$th iteration for each $t \in \{0, \ldots, T-1\}$ is the calculation of an unbiased estimate $\hat{g}^{(t)}$ of the gradient $\nabla I$, where $I$ is given by (103). The gradient of $I$ is given by

$$
\begin{aligned}
\nabla I(\alpha) &= \sum_{\tilde{x} \in \tilde{\mathcal{X}}} p^{(\rho)}(\tilde{x}) \begin{pmatrix} 2\Re\left[\mathrm{e}^{-2\pi i v_0 \cdot \tilde{x}}\left(f(\tilde{x}) - \sum_{m=0}^{M-1} \alpha_m \mathrm{e}^{2\pi i v_m \cdot \tilde{x}}\right)\right] \\ \vdots \\ 2\Re\left[\mathrm{e}^{-2\pi i v_{M-1} \cdot \tilde{x}}\left(f(\tilde{x}) - \sum_{m=0}^{M-1} \alpha_m \mathrm{e}^{2\pi i v_m \cdot \tilde{x}}\right)\right] \end{pmatrix} \\
&= \sum_{m=0}^{M-1} \frac{1}{M} \sum_{\tilde{x} \in \tilde{\mathcal{X}}} p^{(\rho)}(\tilde{x}) \begin{pmatrix} 2\Re\left[\mathrm{e}^{-2\pi i v_0 \cdot \tilde{x}}\left(f(\tilde{x}) - M\alpha_m \mathrm{e}^{2\pi i v_m \cdot \tilde{x}}\right)\right] \\ \vdots \\ 2\Re\left[\mathrm{e}^{-2\pi i v_{M-1} \cdot \tilde{x}}\left(f(\tilde{x}) - M\alpha_m \mathrm{e}^{2\pi i v_m \cdot \tilde{x}}\right)\right] \end{pmatrix},
\end{aligned} \tag{108}
$$

where $\Re$ represents the real part. In the $t$th iteration, Algorithm 3 estimates the gradient at a point denoted by

$$\alpha^{(t)} = \begin{pmatrix} \alpha_0^{(t)} \\ \vdots \\ \alpha_{M-1}^{(t)} \end{pmatrix} \in \left\{ \alpha^{(1)}, \dots, \alpha^{(T)} \right\}. \tag{109}$$

Using a pair of given data points $(\tilde{x}_t, y_t = f(\tilde{x}_t)) \in \{(\tilde{x}_0, y_0), (\tilde{x}_1, y_1), \dots, \}$ sampled with probability $p^{(\rho)}(\tilde{x})$ as observations of an independently and identically distributed (IID) random variable, and an integer $m \in \{0, \dots, M-1\}$ uniformly sampled with probability $\frac{1}{M}$, we give an unbiased estimate $\hat{g}^{(t)}$ of this gradient at each point $\alpha^{(t)}$ by

$$\hat{g}^{(t)} = \begin{pmatrix} 2\Re \left[ e^{-2\pi i v_0 \cdot \tilde{x}_t} \left( y_t - M\alpha_m^{(t)} e^{2\pi i v_m \cdot \tilde{x}_t} \right) \right] \\ \vdots \\ 2\Re \left[ e^{-2\pi i v_{M-1} \cdot \tilde{x}_t} \left( y_t - M\alpha_m^{(t)} e^{2\pi i v_m \cdot \tilde{x}_t} \right) \right] \end{pmatrix}. \tag{110}$$

By construction, we have

$$\mathbb{E} \left[ \hat{g}^{(t)} \right] = \nabla I \left( \alpha^{(t)} \right). \tag{111}$$

We obtain $\tilde{x}_t$ using the classical oracle $\mathcal{O}_{\tilde{x}}$ within time $T_{\tilde{x}}$, and $y_t = f(\tilde{x}_t)$ using the classical oracle $\mathcal{O}_y$ within time $T_y$. As for $m$, since we can represent the integer $m$ using $\lceil \log_2(M) \rceil$ bits, where $\lceil x \rceil$ is the least integer greater than or equal to $x$, we can sample $m$ from a uniform distribution using a numerical library for generating a random number within time $O(\text{polylog}(M))$. Note that even in case it is expensive to use randomness in classical computation, quantum computation can efficiently sample $m$ of $\lceil \log_2(M) \rceil$ bits from the uniform distribution within time $O(\log(M))$. In this quantum algorithm, $\lceil \log_2(M) \rceil$ qubits are initially prepared in $|0\rangle^{\otimes \lceil \log_2(M) \rceil}$, and the Hadamard gate $H = \frac{1}{\sqrt{2}} \begin{pmatrix} 1 & 1 \\ 1 & -1 \end{pmatrix}$ is applied to each qubit to obtain

$$\frac{1}{\sqrt{2^{\lceil \log_2(M) \rceil}}} (|0\rangle + |1\rangle)^{\otimes \lceil \log_2(M) \rceil}, \tag{112}$$

followed by a measurement of this state in the computational basis to obtain a $\lceil \log_2(M) \rceil$-bit outcome sampled from the uniform distribution. Given $\tilde{x}_t$, $y_t$, and $m$, we can calculate each of the $M$ element of $\hat{g}$ in (110) within time $O(D)$ for calculating the inner product of $D$-dimensional vectors, and hence the calculation of all the $M$ elements takes time $O(MD)$. Therefore, each iteration takes time

$$O(T_{\tilde{x}} + T_y + \text{polylog}(M) + MD) = O(MD + T_{\tilde{x}} + T_y). \tag{113}$$

Note that without sampling $m$, we would need $O(M^2 D)$ runtime per iteration because each of the $M$ elements of the gradient in (108) includes the sum over $M$ terms; thus, the sampling of $m$ is crucial for achieving our $O(MD)$ runtime.

To bound the required number of iterations, we use an upper bound of the number of iterations in Algorithm 3 given in Ref. [24], which shows that if we have the following:

- for any $\alpha \in \mathcal{W}$,

$$\|\nabla I(\alpha)\|_2 \leqq L, \tag{114}$$

- the unbiased estimate $\hat{g}$ for any point $\alpha \in \mathcal{W}$ almost surely satisfies

$$\|\hat{g}\|_2 \leqq L, \tag{115}$$

- the diameter of $\mathcal{W}$ is bounded by

$$\text{diam}\,\mathcal{W} \leqq d, \tag{116}$$

then, after $T$ iterations, with high probability greater than $1 - \delta$, Algorithm 3 returns $\alpha$ satisfying

$$\epsilon = O\left( dL \sqrt{\frac{\log\left(\frac{1}{\delta}\right)}{T}} \right), \tag{117}$$

where we write
$$\epsilon = I(\alpha) - \min_{\alpha \in \mathcal{W}} \{I(\alpha)\}. \tag{118}$$

In the following, we bound $d$ and $L$ in (117) to clarify the upper bound of the required number of iterations $T$ in our setting.

To show a bound of $d$, recall the assumption that we are given a sufficiently large number $M$ of features for achieving the learning in our setting. Then, Bach [25] has shown that with the $M$ features sampled from the weighted probability distribution $Q(v_m)P^{(\tau)}(v_m)$ by Algorithm 1, the learning to the accuracy $O(\epsilon)$ can be achieved with coefficients satisfying

$$\|\beta\|_2^2 = O\left(\frac{1}{M}\right), \tag{119}$$

where $\beta = (\beta_0, \ldots, \beta_{M-1})^{\mathrm{T}}$ is given for each $m$ by

$$\beta_m = \sqrt{Q(v_m)}\alpha_m. \tag{120}$$

This bound yields

$$\sum_{m=0}^{M-1} Q(v_m)\alpha_m^2 = O\left(\frac{1}{M}\right). \tag{121}$$

In the worst case, a lower bound of the left-hand side is

$$\sum_{m=0}^{M-1} Q(v_m)\alpha_m^2 \geqq Q_{\min} \|\alpha\|_2^2. \tag{122}$$

From (121) and (122), we obtain an upper bound of the norm of $\alpha$ minimizing $I$

$$\|\alpha\|_2^2 = O\left(\frac{1}{MQ_{\min}}\right). \tag{123}$$

Thus, it suffices to choose the parameter region $\mathcal{W}$ of $\alpha$ as an $M$-dimensional ball of center 0 and of radius $O\left(\frac{1}{\sqrt{MQ_{\min}}}\right)$, which yields the diameter

$$d = O\left(\frac{1}{\sqrt{MQ_{\min}}}\right). \tag{124}$$

As for a bound of $L$, we obtain from (110) and (123)

$$\|\hat{g}\|_2 = O\left(M \|\alpha\|_2 + \sqrt{M}\right) = O\left(\sqrt{\frac{M}{Q_{\min}}} + \sqrt{M}\right) = O\left(\sqrt{\frac{M}{Q_{\min}}}\right), \tag{125}$$

where we take the worst case of small $Q_{\min}$, and we use bounds

$$\sqrt{\sum_{m=0}^{M-1} |M\alpha_m \mathrm{e}^{2\pi i v_m \cdot \tilde{x}_t}|^2} = O(M\|\alpha\|_2), \tag{126}$$

$$\sqrt{\sum_{m=0}^{M-1} y_t^2} = O(\sqrt{M}). \tag{127}$$

Since the upper bound of $\|\hat{g}\|_2$ is larger than $\|\nabla I(\alpha)\|_2$, we have

$$L = O\left(\sqrt{\frac{M}{Q_{\min}}}\right). \tag{128}$$

Therefore, using (124) and (128), we bound the right-hand side of (117)

$$\epsilon = O\left(dL\sqrt{\frac{\log\left(\frac{1}{\delta}\right)}{T}}\right) = O\left(\frac{1}{Q_{\min}}\sqrt{\frac{\log\left(\frac{1}{\delta}\right)}{T}}\right). \tag{129}$$

Thus, it follows that

$$T = O\left(\frac{1}{\epsilon^2 Q_{\min}^2} \log\left(\frac{1}{\delta}\right)\right). \tag{130}$$

Combining (107), (113), and (130), we obtain the claimed overall runtime. □