[Reviews · NeurIPS 2020]

Review 1

Summary and Contributions: The paper present a quantum algorithm for sampling Fourier features with probabilities proportional to the ridge leverage scores of the kernel. The proposed algorithm can generate a sample from this distribution in time nearly proportional to the dimension of the dataset.

Strengths: Fast quantum algorithm for large scale kernel method. Theoretical analysis of the proposed method.

Weaknesses: The paper introduces quantization of the data and Fourier domain but does not analyze the error that is introduced by this. In particular, in page 5, the data domain is quantized and the points of the dataset are rounded to their nearest grid point, however, the effect of this rounding is not quantified. This can be an issue. For instance, if a dataset contains a cluster of points that are all very close to each other such that after rounding they will be all rounded to the same grid point. In this case even if you compute the exact kernel matrix after rounding, it will not be spectrally close to the original kernel matrix.

Correctness: Besides the rounding issue, I did not find any issues in the proof.

Clarity: Yes the paper is mostly clear.

Relation to Prior Work: I do not agree with the claim that is made about the prior work of [20]. The authors criticize [20] for not sampling the Fourier features with respect to the ridge leverage scores distribution of the integral operator \Sigma. First of all, this integral operator in unknown in practice because it is defined by making the assumption that the distribution from which the dataset is generated in known. This assumption is clearly not true in most real world settings. In particular, in page 5 the authors try to circumvent the issue of having unknown distribution on the input dataset by approximating the distribution using a heuristic method whose performance is not analyzed in the paper. Second, the focus of [20] is to spectrally preserve the kernel matrix K whose entries are the kernel value on pairs of points from the input dataset. This is sufficient for proving that the solution to any of the downstream kernel methods obtained by using the approximated kernel matrix is a (1+\eps) approximation to solution of the exact kernel method. The spectral approximation to the kernel matrix K is obtained in [20] by sampling the Fourier features with respect to the leverage scores of the kernel matrix K.

Reproducibility: Yes

Additional Feedback: See the above sections "Relation to prior work" and "Weaknesses". ============================================== I thank the authors for taking the time to provide feedback. I find the clarifications in the rebuttal very helpful. My assessment is that the presentation of the paper is suboptimal and it is hard to see the main techniques/ideas. The authors have addressed my concerns about rounding and also the position of this work in relation to [20]. I encourage the authors to add these clarifications to the main body of the paper. Given this, I decided to raise my score.


Review 2

Summary and Contributions: This paper proposed a quantum algorithm for kernel methods with random features. Specifically, their quantum algorithm runs in time O(D) where D is the dimension of the input data, whereas classical algorithms take O(exp(D)) time.

Strengths: I think the main strength of this paper is the claim that it does not have sparsity or low-rank assumptions. These two assumptions have been dominant in previous literature on quantum machine learning, and it’s of general interest to remove such assumptions which can be too strong in practice. Roughly speaking, this paper achieved this using the quantum Fourier transform, the core behind the Shor’s algorithm for integer factorization.

Weaknesses: I think the main weakness of this paper is that the writing is too dense to parse, while on the other hand the current content in the main body is not enough to examine the correctness of Theorem 1 and 2. To be more specific, Theorem 1 and 2 are all both technical results, especially Theorem 1. From my understanding of Section 3.2 and the relevant part in the appendices, the authors used the quantum RAM as well as the quantum singular value transformation (QSVT) algorithm to achieve the speedup for sampling from the features using the quantum Fourier transform. I understand that NeurIPS submissions have page limitation, but I think all the steps should at least be highlighted. In particular, I feel that discussions are needed for: - What quantum RAM do we need for the task? - What can we use the QSVT without the sparse or low-rank assumption? They are actually important factors to turn into the block encoding (see Lemma 48-50 of https://arxiv.org/pdf/1806.01838.pdf) - How does the quantum Fourier transform come into the overall algorithm? On the other hand, I’m not totally sure how Section 2.3 (discretized representation of real number) and Section 2.4 (assumption on data in discretized representation) help with the overall story--I can foresee their usage, but the paper can probably shrink this space and highlight more on the proofs of Theorem 1 and 2. These are the two main technical results, but are only given in the last page (Page 8) without a comprehensive discussion. A practical solution is to fulfill more details between Line 251-272. In all, my feeling of the current version of the paper is that it somehow manages to show a bound that’s independent of sparsity or low-rankness in an explicit way, but the problem is involved, the algorithm has various of black-boxes, and I’m not totally sure that after opening all the black-boxes between the connections among QRAM, QSVT and quantum Fourier transform, the claim is as strong as the current one. Minor points: - This paper didn’t give any discussion on the practical side of their quantum algorithm. It is fine as a theory paper, but for the general interest of the NeurIPS, it would be much better to introduce more of that. For instance, the paper [47] by Havlicek et al. applied the kernel method for a supervised learning problem on a 5-qubit machine. Although Ref. [47] has drawbacks in precision, but considering the similarity between methods and problems, and for the interest of the implementation side, the current submission should probably discuss more. - I’m a bit confused about Line 59-63. It seems to me that if the inverse operator is hard to calculate, it is purely a mathematical thing, and requiring exp(D) dimension should apply to both classical and quantum. What’s the difference here?

Correctness: The overall story looks plausible, but as I said in the weakness part, it’s not easy to verify the correctness of Theorem 1 and 2 without checking many details of the supplementary material. This paper is purely theoretical, so the second question doesn’t apply.

Clarity: The English is fine, but the presentation is too dense, and more efforts can probably be made to articulate the algorithms and the proofs behind.

Relation to Prior Work: Yes, as far as I see. A minor point: some reference information was outdated. Ref. [41] was accepted by STOC 2020 and Ref. [40] was accepted by MFCS 2020.

Reproducibility: Yes

Additional Feedback: After rebuttal: I would like to thank the authors very much for answering my questions as well as those from other reviewers'. I'm happy to increase my score. The main reason is that the original presentation didn't articulate the technical contributions of the work; after having explicit pointers and discussions, I have a better understanding about the paper, and it turns out that the contribuion is more significant than I previously thought. Another minor comment I just realized: There seems to be minor style issues of the references -- all the titles of the papers are omitted. This is not very common in computer science papers; is it possible to fix this?


Review 3

Summary and Contributions: This paper considers feature extraction using quantum machine learning (QML) algorithms. In fact, it is using classical mathematical tools to analyze the QML algorithm running time to speed up the random feature extraction. The claimed contributions are the algorithm is speed up without considering sparsity and low-rank assumptions. Especially, the running time of O(D) is attractive but might not be feasible.

Strengths: The paper is well written. It describes the problem well, though there is a long introduction before the contribution of this paper. But it might be necessary for a topic of QML. +) Motivation of this work is clear and strong. +) The derivation of the model from (5) to (8) is solid. I assume (8) is actually the real formulation and the starting point of the contribution of this paper.

Weaknesses: My concerns lie in the following aspects: -) The contributions of this paper is actually 2 parts: 1) the O(D) running time and 2) without considering the sparsity and low-rank. However, for 1) in line 270, I don't think the \tilde O(Q_max / polylog(1/\Delta)) can be ignored. Considering the Delta is very small, then this term can be big due to 1/\Delta. In this case, the whole statement of O(D) does not hold anymore. For 2), even the authors do not consider sparsity or low rank, the proof seems to me using concentration-of-measure, i.e, the high probability considering the tails of the distribution. In this case, it essentially has some connection to low rank. So the claim of not considering low rank is not that strong either. -) Even the claims are true (most likely not), the results have been in [39][42][44] It seems the result is a simple extension of existing results.

Correctness: As mentioned in the weakness, it seems the claims are not well defined. I am not an expert in QML, so I cannot evaluate the applications of these methods.

Clarity: The readability of this paper is hard, especially the reference. No title of the the paper is presented in the reference. This costs too much time for the reviewers. For example, in [7], F. Bach has two papers in JMLR and they are in the same issue. Which one does the authors want to cite?

Relation to Prior Work: Yes. Most of them are clear.

Reproducibility: Yes

Additional Feedback: The contributions should be clearly defined without over claiming. Some real applications might need to be considered. === Post Rebuttal: After read the rebuttal. I am OK with O(D). However, for the "concentration inequality" based proof, it essentially impose sparse or low-rank. The contribution is incremental.

[Author Response · NeurIPS 2020]

We appreciate Reviewers' comments and will improve presentation and reference lists based on the following responses.

(Reviewer 1: The effect of rounding: e.g., a cluster of points.) Rounding may have some effect, but this effect does not ruin the impact of our contributions, i.e., construction of the fast quantum algorithm and evaluation of its runtime. **After all, any implementation of kernel methods by computer with bits requires rounding, and in our setting, we resolve a cluster of points by rescaling, which is equivalent to increasing precision of rounding without rescaling.** As discussed in Sec. 3.1, our rescaling keeps the learning problem invariant. Then under standard assumptions in signal processing where such implementation works well, it should be straightforward to show our algorithm also works well.

(Reviewer 1: Difference from Ref. [20].) Compared to Ref. [20], the significance of our contributions is that **our algorithm in the limit of good approximation (as $N, G \to \infty$) is provably optimal** up to a logarithmic gap as shown in Ref. [7]; also, its runtime is as fast as poly-logarithmic in $N, G$ as shown in Sec. 3.2. Regarding the first point of the review comment, the integral operator may be unknown in practice, but our algorithm converges to sampling from the leverage-score distribution of the integral operator as $N, G \to \infty$ whereas **Ref. [20] does not converge to this optimal distribution in any limit**. As for the second point of the review comment, Ref. [20] can achieve the learning since it preserves the kernel (Gram) matrix, but the optimality of Ref. [20] using the kernel matrix (i.e., a gap from a lower bound of the required number of random features) is unknown in general; in contrast, Ref. [7] using the integral operator proves the optimality, and **our algorithm based on Ref. [7] achieves this optimality** in the limit of $N, G \to \infty$.

(Reviewer 2: What quantum RAM (QRAM) do we need for the task? The algorithm has various black boxes.) We need a quantum oracle $\mathcal{O}_\rho$ defined explicitly in the first paragraph of Sec. 3.2. **This oracle is the only black box in our quantum algorithm; putting effort to make our algorithm explicit, we avoid any other QRAM.** This input model is the same as quantum recommendation systems in Ref. [37] and is implementable feasibly as discussed in Sec. B of Supplemental Material, which we omit from the main text since it is well established in Ref. [37].

(Reviewer 2: What (why) can we use the QSVT without the sparse or low-rank assumption?) We can avoid sparse and low-rank assumptions **because we explicitly decompose the (non-sparse and full-rank) operator $\Sigma_\epsilon$ into addition and the multiplication of diagonal (i.e., sparse) operators and QFTs**, so that we can construct an efficient implementation of the block encoding of $\Sigma_\epsilon$. This main technical contribution of our paper is explained in the last paragraph of Sec. 3.1. Remarkably, our technique does not directly use Lemmas 48–50 of arXiv:1806.01838 since these lemmas require sparse and low-rank assumptions. While we could implement $\Sigma_\epsilon$ by addition and multiplication of block encodings of the diagonal operators and QFTs, the presentation of these additions and multiplications may become complicated since we have multiple block encodings to be combined. For simplicity of the presentation, we use the block encoding of the POVM operator (Lemma 46 of arXiv:1806.01838) at the technical level to represent how to combine all the block encodings and QFTs as one circuit, as shown in Figs. 1 and 2 of Supplemental Material.

(Reviewer 2: How does the quantum Fourier transform come into the overall algorithm?) Our algorithm uses QFTs **for implementing the block encoding of $\Sigma_\epsilon$ and for applying $\mathbf{F}_D^\dagger$ in preparing the quantum state (14).**

(Reviewer 2: On the practical side of the quantum algorithm.) Rather than 5 qubits in Ref. [47], our algorithm aims at applications at large scales. In contrast to Ref. [47], we prove that our algorithm achieves the exponential speedup. Thus, as explained in Introduction, **our algorithm is a convincing candidate for killer applications of universal quantum computers in the long run; after all, large-scale machine learning will be eventually needed in practice.**

(Reviewer 2: What's the difference between classical and quantum?) The existing classical algorithm calculates description of probability distribution by matrix inversion, and then perform sampling. In contrast, **our quantum algorithm does not estimate the classical description of the distribution represented by the amplitude of quantum states.**

(Reviewer 3: Considering $\Delta$ is very small, then $\mathrm{polylog}(1/\Delta)$ can be big, and $O(D)$ does not hold anymore.) **The precision factor $\mathrm{polylog}(1/\Delta)$ is ignorable in practice** while we explicitly write it for correctness of our runtime analysis. For example, consider two $D$-dimensional real vectors $x$ and $y$. Computers with bits can use fixed-point number representation to represent each real element with precision $\Delta$ using $O(\log(1/\Delta))$ bits (e.g., 64 bits). In this case, multiplication of two elements requires $O(\mathrm{polylog}(1/\Delta))$ runtime, and calculation of inner product of $x$ and $y$ requires $O(D\,\mathrm{polylog}(1/\Delta))$ runtime, but the factor $\mathrm{polylog}(1/\Delta)$ is practically ignored.

(Reviewer 3: It essentially has some connection to low rank.) Using a concentration inequality, Ref. [7] shows that a requirement for any algorithm using random features to achieve the learning with reasonable runtime and accuracy is given in terms of the degree of freedom, in particular, by the bound (5) in our paper. However, **this requirement (5) does not imply low rank of operators used in our algorithm,** as discussed in the second paragraph of Sec. 3.2.

(Reviewer 3: The results have been in [39][42][44]) The novelty of our contributions is that we construct an exponentially faster QML algorithm that is free from sparsity and low-rank assumptions. As discussed in Introduction, **this has been challenging but crucial in QML, and none of Refs. [39] [42] [44] achieves this.** We hope that the above explanations about all the questions help with eliminating the concerns about validity and importance of our results.

[Meta-Review · NeurIPS 2020]

The paper gives an O(input dimension) quantum ML algorithm for kernel methods with random Fourier features, without requiring sparsity or low-rank assumptions. This is a much improved resubmission from a different venue, and is able to avoid relying on quantum RAM. The fact that it relies on only standard input and does not rely on assumptions on the data are very nice. Some of the reviewers were not as convinced looking at the classical techniques (RFF), but after discussion they ultimately agreed that this is a significant contribution to Quantum ML. However there are a few clarity issues that will need to be addressed in the final version.